# Direct observation of deformation and resistance to damage accumulation during shock loading of stabilized nanocrystalline Cu-Ta alloys

B. C. Hornbuckle [1] ✉, R. K. Koju [2], G. Kennedy[3], P. Jannotti[1], N. Lorenzo[1], J. T. Lloyd[1], A. Giri[1], K. Solanki [4], N. N. Thadhani[3], Y. Mishin [2] & K. A. Darling[1]

Energy absorption by matter is fundamental to natural and man-made processes. However, despite this ubiquity, developing materials capable of withstanding severe energy fluxes without degradation is a significant challenge in materials science and engineering. Despite recent advances in creating alloys resistant to energy fluxes, mitigating the damage caused by the absorption and transfer of mechanical energy remains a critical obstacle in both fundamental science and technological applications. This challenge is especially prominent when the mechanical energy is transferred to the material by shock loading. This study demonstrates a phenomenon in which microstructurally stabilized nanocrystalline Cu-Ta alloys can undergo reversal or nearly complete recovery of the dislocation structure after multiple shock-loading impacts, unlike any other known metallic material. The microstructure of these alloys can withstand repeated shock-wave interactions at pressures up to 12 GPa without any significant microstructural damage or deterioration, demonstrating an extraordinary capacity to be virtually immune to the detrimental effects of shock loading.

Developing materials that can withstand severe energy fluxes remains one of the grand challenges in materials science[1–3]. The absorption of high-energy fluxes, such as intense heat, harsh irradiation, or severe mechanical energy transfer, can cause structural degradation or even destruction of most materials[4,5]. This degradation occurs as crystallographic defects proliferate, causing the accumulation of structural damage and eventual failure[4,5]. Exploration and discovery of new materials capable of enduring high-energy fluxes are required to address numerous scientific and technological challenges[1–3,6–9]. Despite previous efforts[10–12], progress in developing materials immune to mechanical energy absorption or transfer has been limited, especially under high pressure caused by shock deformation.

Plastic deformation, which is a manifestation of the dissipating mechanical energy, is path-dependent and not a state function[13]. It is affected by the type, magnitude, and history of the applied energy flux, which determines the distribution and overall concentration of defects formed. If two samples were subjected to the same total strain but one was deformed by a dynamic load while the other quasi-statically, they would exhibit different microstructures[14] with distinct internal defect structures. In particular, when subjected to shock loading to a high-pressure state on the Hugoniot (the locus of shock pressure states), the microstructure of most polycrystalline metals undergoes a breakdown, transitioning into a highly defective non-equilibrium state[4,5]. This state is characterized by a high concentration of vacancies,

[1]Army Research Directorate, DEVCOM Army Research Laboratory, APG, MD 21005, USA. [2]Department of Physics and Astronomy, George Mason University, MSN 3F3, Fairfax, VA 22030, USA. [3]School of Materials Science and Engineering, Georgia Institute of Technology, 771 Ferst Dr. NW, Atlanta, GA 30332, USA. [4]School for the Engineering of Matter, Transport, and Energy, Arizona State University, Tempe, AZ 85287, USA. ✉e-mail: billy.c.hornbuckle.civ@army.mil

interstitials, dislocations, stacking faults, twins/microbands, cracks, and voids. Many examples of the drastic changes in Cu and other face-centered-cubic (FCC) metals caused by shock loading can be found in the literature[4,5,9,15,16]. These changes often render the original microstructure unrecognizable[4,5,9,15,16]. The detrimental consequences of such damage are amplified when metals undergo repeated shock loading events[9,17], leading to the accumulation of additional damage. These adverse effects cause various issues, such as incipient and catastrophic spall failure and significant embrittlement by shock hardening[5,18].

This study presents a phenomenon observed in stabilized nanocrystalline (NC) Cu-Ta alloys. This phenomenon is the energy dissipation and nearly full recovery from the damage created by shock loading to high pressures, even after multiple loading events. This response to shock loading contradicts what is known about shock loading for all other metallic materials. Here, we have studied NC Cu alloys containing 1 and 3 at.% Ta (abbreviated as Cu-1Ta and Cu-3Ta, respectively). In comparison, ultrafine-grained (UFG) oxygen-free high-conductivity (OFHC) Cu obtained by equal channel angular extrusion (ECAE) has also been studied. The shock deformation of these materials was performed using the traditional gas gun flyer plate impact method. The samples were characterized by Scanning Transmission Electron Microscopy (STEM) and X-ray diffraction (XRD), including the time-resolved capabilities at the Advanced Photon Source (APS). Initial microstructures of these alloys have been described in the Methods and the previous literature[19-24].

It is known that the high-temperature thermo-mechanical stability of the NC Cu-Ta alloys increases with the Ta concentration[11,19,20,22,24-26]. The mechanism of the nano-stabilization has been investigated in detail[11,19,20,22,24-28]. Cu and Ta are immiscible in the solid state, but mechanical alloying by ball milling forces a significant amount of Ta into a metastable Cu-Ta solid solution. The subsequent annealing at moderate temperatures initiates a decomposition of the solution accompanied by precipitation of a high density of coherent Ta clusters

distributed inside the nano-grains and at grain boundaries. These clusters pin the grain boundaries by the Zener pinning mechanism[27,28], effectively suppressing the grain growth and stabilizing the nanostructure. The degree of stabilization depends on the cluster density, which in turn depends on the overall Ta content. Prior research has determined that the highest stability is achieved by the Cu-3Ta composition[20,22]. Hence, this alloy was chosen for the present study. The NC Cu-1Ta alloy is known to be less stable[19,20] and was included here to show the impact of structural stabilization on the shock response. The results reported below demonstrate the ability of the Cu-Ta alloys to absorb damage without significant changes in the grain size or mechanical properties. The microstructural components of these alloys exhibit exceptional resilience when subjected to repeated shock loadings at pressures up to 12 GPa, demonstrating a remarkable ability to remain virtually unaffected by the transfer of intense mechanical energy.

## Results

### Microstructural evolution under shock loading

After repeated shock loading at a pressure of 12 GPa (calculated using two-dimensional simulations), the samples were soft-recovered, and their microstructure was examined to assess their resilience to shock deformation. Information related to the pressure-time and velocity histories during repeated shock recovery experiments and in-situ measurements using XRD of the materials studied herein is presented in Supplementary Fig. 1 and 2. For the OFHC copper used here, for comparison, prior research has documented extensive stress-induced grain growth[29]. This phenomenon occurs when the microstructure's average grain size is in the NC and UFG range, mirroring observations found in other metals[30-32]. Such extensive coarsening has been reported in all known deformation modes, including tension, compression, fatigue, indentation, and creep[25,29,33-39]. This stress-induced grain growth reflects the lack of microstructural stability of NC and UFG metals, including Cu. This instability was further confirmed in our work. Supplementary Fig. 3 shows the grain orientation map obtained by Electron Backscattered Diffraction (EBSD) imaging. The map reveals a significant (over 500%) increase in grain size near scratches introduced during the final stages of sample polishing. The stress effect on the microstructure was more dramatic in the Cu samples subjected to shock loading, as demonstrated in Fig. 1. The micrographs in Fig. 1 compare the microstructures before (Fig. 1A) and after (Fig. 1B) one cycle of shock loading. The shock deformation has completely altered the microstructure, causing extensive grain growth (quantified later) and a high level of damage accumulation inside the grains. The damage manifests itself in the contrast variations, which are due to the large residual strain energy density stored within the microstructure.

To further investigate the response of Cu to the shock deformation, high-magnification STEM imaging was performed. Supplementary Figs. 4C and 5C show ultra-high-resolution (11 k x 15 k pixels) STEM images over a 50 micron square of the electron-transparent microstructure of Cu after the first and second rounds of shock loading. As expected and consistent with the literature[5], the post-shock microstructure contains a large amount of damage accumulated in the form of dislocations arranged in dense cell structures. The high dislocation density regions are distributed throughout the microstructure, indicating a significant amount of the mechanical energy was captured and stored as defects within the material. In addition to the STEM analysis, EBSD was performed on the impact and rear faces of the sample and from the middle of the sample. The EBSD image shown in Supplementary Fig. 6 quantifies the shock-induced grain growth. The average grain size increases from 350 nm to ~25 micrometers (a factor of 60–70 increase).

In contrast to Cu, Fig. 1C–E and Supplementary Figs. 4 and 5 show that the microstructure of both NC Cu-Ta alloys exhibits minimal to no change after the shock deformation despite their considerably smaller

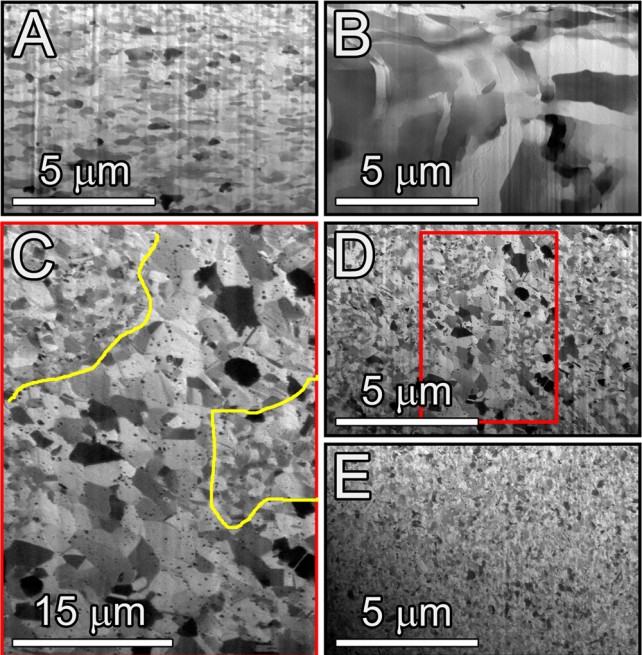

**Fig. 1 | Ion contrast image showing the grain structure of Cu and Cu-Ta alloys.** **A**, **B** Cu before (**A**) and after (**B**) shock compression at ~12 GPa. **C**, **D** Cu-1Ta and (**E**) Cu-3Ta alloys after shock compression at ~12 GPa. **C** Higher magnification image of the region within the red box in (**D**) outlines regions within the Cu-1Ta alloy that experienced some degree of abnormal grain growth. Yellow lines delineate abnormally grown grains from regions of nanocrystalline grains.

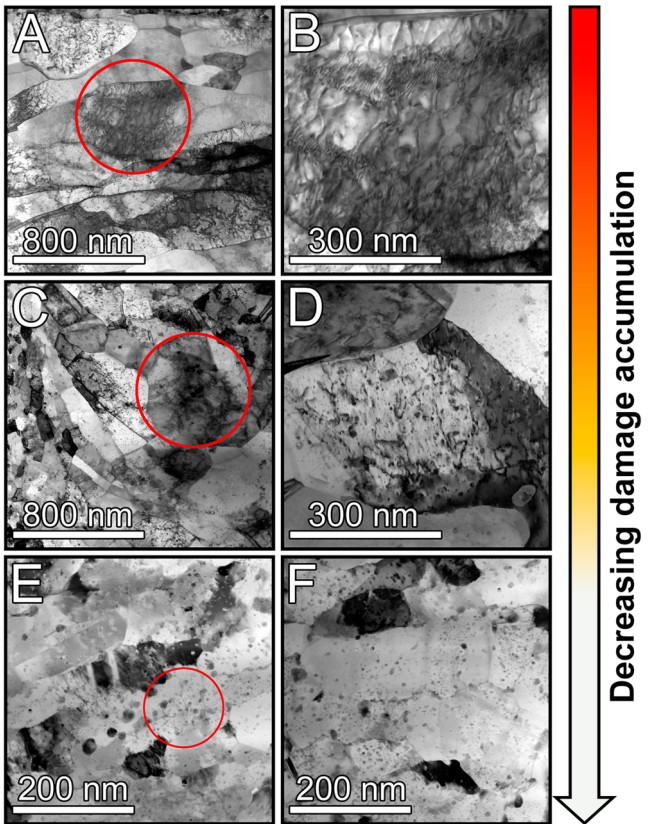

**Fig. 2 | Medium magnification STEM bright-field images. A**, **B** Cu; **C**, **D** Cu-1Ta; and **E**, **F** Cu-3Ta after the first 12 GPa shock compression. Red circles show regions of damage and dislocation activity.

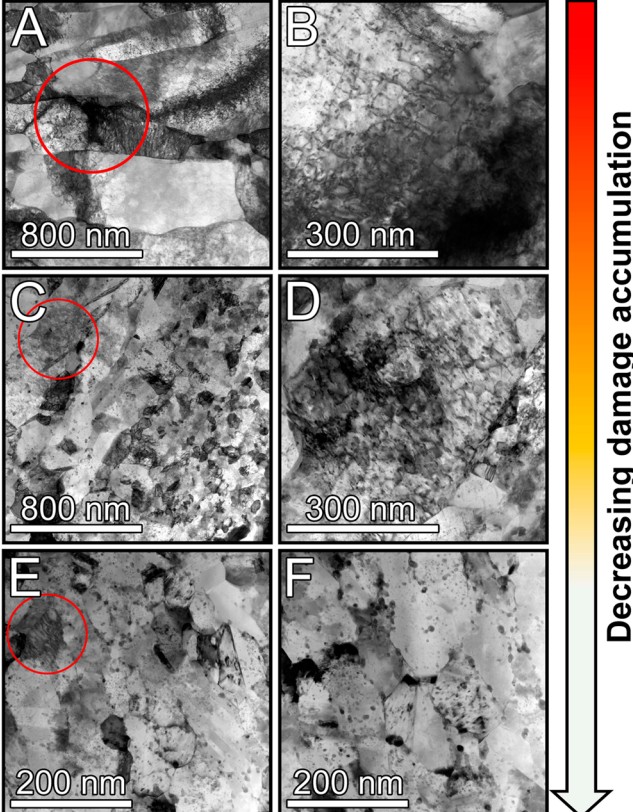

**Fig. 3 | Medium magnification STEM bright-field images. A**, **B** Cu; **C**, **D** Cu-1Ta; and **E**, **F** Cu-3Ta after the second 12 GPa shock compression. Red circles show regions of damage and dislocation activity.

initial grain size (Supplementary Fig. 7). Although some abnormal grain growth occurs in Cu-1Ta, significant regions of the nano-grains remain intact. The Cu-3Ta alloy demonstrates better microstructural stability, exhibiting a more homogeneous post-shock microstructure with a smaller average grain size. This observation is significant because smaller grain sizes create a higher thermodynamic driving force for grain growth. Thus, one would expect that external stimuli, such as a shock wave, would trigger much more rapid grain growth in NC Cu-Ta than in UFG Cu. Remarkably, our experiments reveal the opposite effect.

Figures 2 and 3 present more details of the alloy microstructure after the first (Fig. 2) and second (Fig. 3) shocks. The images show that the microstructural stability increases with the addition of Ta. In Cu-1Ta, there are regions of abnormal grain growth, and it is only within those specific regions (shown in Figs. 2C, D 3C, D) that stored dislocation networks are observed. Because Cu-3Ta's grains are small, we can barely discern individual dislocations (seen in the red circles of Figs. 2E and 3E), let alone dislocation cell structures. This is understandable because dislocation sources cannot be easily activated within small grains[40]. The grains that remain NC are nearly dislocation-free, and the volume fraction of the smaller nano-grains increases as the Ta concentration increases to 3 at%. The nearly defect-free microstructure in Cu-3Ta persists even after the second shock loading (Fig. 3E, F and Supplementary Fig. 8 and 9). In contrast, in Cu and Cu-1Ta, the second shock loading increases residual damage. Supplementary Fig. 8 shows ion contrast images taken from the impact and rear surfaces and the middle part of the Cu-3Ta sample. Supplementary Fig. 9 presents high-magnification STEM bright field images of the nearly defect-free structures at these three locations in the Cu-3Ta, while Supplementary Fig. 10 provides ultra-high-resolution STEM images from the three locations. All these images confirm that the

microstructural stability persists homogeneously throughout the Cu-3Ta samples.

Conventional XRD analysis of the pre-shock and repeat shock-compressed soft-recovered samples shows nearly identical profiles for Cu-3Ta (Fig. 4). Specifically, the peaks have very close widths, positions along the 2θ axis, and relative peak intensities. The absence of changes in the relative peak intensity suggests no alteration in the overall crystallographic orientation of the grain lattices, indicating that the shock loading did not cause a textural change[41]. Additionally, no significant increase in the formation of stacking faults, twin density, or microbands is detected from the STEM analysis. By comparison, when subjected to post-shock loading, Cu-1Ta exhibits a slight decrease in the maximum intensity and an observable increase in the full width at half maximum (FWHM). This slight broadening can be attributed to lattice strain occurring within the abnormally grown grains localized in some areas of the microstructure[41].

## Changes in mechanical behavior after shock loading

The results presented above contradict several decades of shock recovery experiments, wherein the structure-property relationships of various coarse-grained materials (encompassing all crystallographic forms of metals) have been extensively investigated[5,14,15] and found to contain a high density of non-equilibrium defects[5,9,15]. Proliferation of these defects with the square root of the peak shock pressure has been consistently demonstrated and represented by an empirical linear relation with the residual hardness[5,18]. This correlation can be explained by the known constitutive relationship between the strain rate and dislocation density and velocity, or alternatively, between the dislocation nucleation rate and average glide distance. The peak shock pressure significantly increases the dislocation density[42,43] and, thus, the post-shock yield stress and the residual hardness.

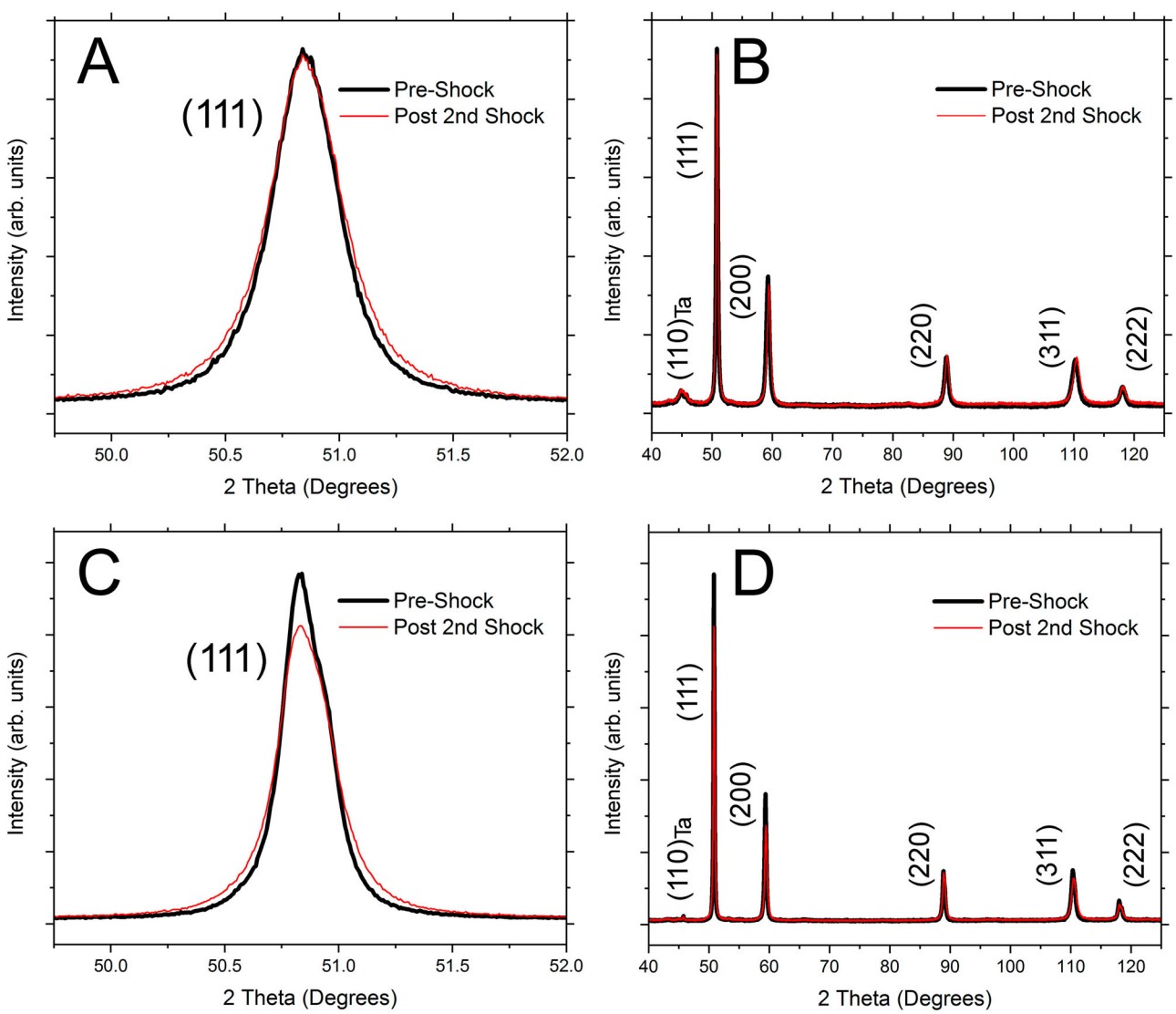

**Fig. 4 | X-ray diffraction patterns for the pre-shock and post-shock compression after the second loading event. A**, **B** NC Cu-3Ta and **C**, **D** NC Cu-1Ta. Source data are provided as a Source Data file.

Literature examples showing the residual hardness for various types of metals and alloys shock-compressed to ~ 15 GPa are provided in Table 1[5]. The hardness values for Cu-1Ta and Cu are not included because these materials undergo noticeable microstructural changes (such as grain growth and increased dislocation accumulation). As a result, it is more challenging to separate the loss of hardness due to grain growth from the increase in hardness caused by the increased dislocation density. For conventional coarse-grain metals, the residual

hardness is in the range of 1.5–2.4 times the initial unshocked value. This is in large contrast to Cu-3Ta, wherein the initial hardness of $304 \pm 5.8$ VHN only changes to $293 \pm 4.4$ VHN after repeated shock loading events. This slight (3.6%) change is within the margin of measurement error, suggesting that the hardness remains virtually unchanged. This result aligns with the absence of any significant accumulation of defects in the microstructure of this alloy. The through-thickness hardness profile for Cu-3Ta can be seen in Supplemental Fig. 11.

### In-situ shock experiments

A series of shock loading experiments were carried out at the Dynamic Compression Sector of Argonne National Laboratories' APS[44] to confirm that the absence of residual dislocations in the Cu-Ta samples was not due to post-shock reversal of bowed dislocations. Snapshots of integrated X-ray diffraction patterns were extracted at four distinct time points, with intervals of 153.4 ns, following the impact. Finite element simulations of the impact event, shown in Supplementary Fig. 2, were used to determine the stress history of the sample compared to the timing of X-rays. The complete X-ray diffraction patterns data can be found in Supplementary Fig. 12.

Snapshots of XRD profiles at various time points provide the kinetics of the absorption phenomena, as illustrated in Fig. 5. The black

### Table 1 | Various metals and alloys shock compressed to 15 GPa

| Metal | Residual hardness change factor (x) |
|---|---|
| Cu | 2.4 |
| Brass | 2.4 |
| Ni | 1.8 |
| 304 SS | 1.5 |
| Inconel 600 | 2.0 |
| Mo | 1.9 |
| NC-Cu-3Ta | 1.0 |

Literature data from[5].

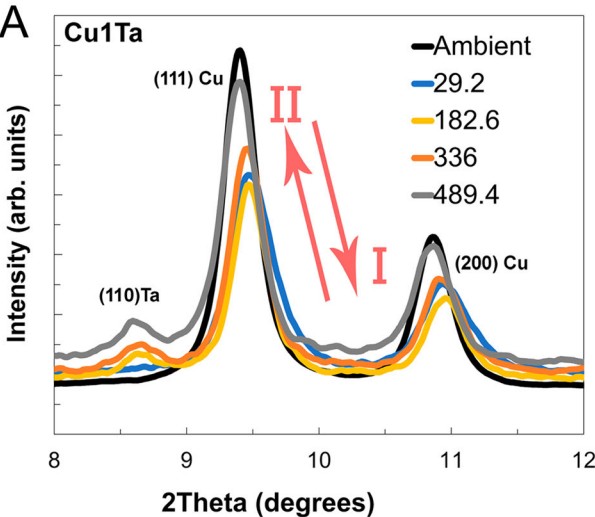

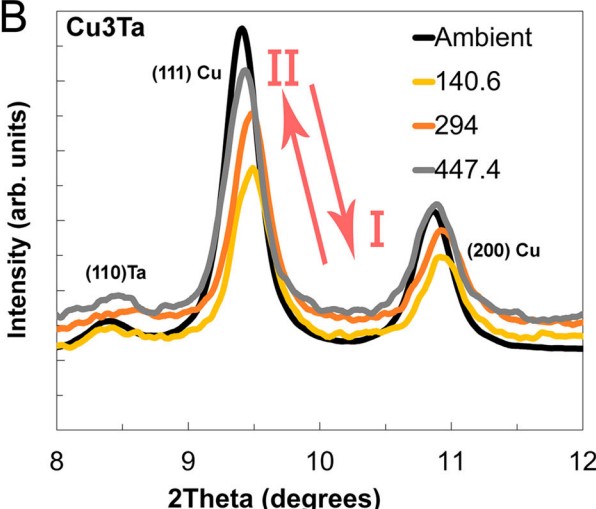

**Fig. 5 | APS data showing the expansion and retraction of the line broadening from diffraction patterns. A** NC Cu-1Ta and **B** NC-Cu-3Ta collected at 153.4 nanosecond time intervals. Source data are provided as a Source Data file.

trace represents the ambient (pre-shock) profile for comparison with the time-resolved XRD data obtained during the loading and unloading stages (as specified in the plot legend). The increasing 2θ position of the peak indicates compression, whereas the peak position close to the ambient value indicates unloading. Peak broadening is observed during incipient deformation (from 1 to 200 ns) for Cu-1Ta and Cu-3Ta, indicated as *I* in Fig. 5. This broadening is a fingerprint of active defect generation, such as dislocation nucleation or multiplication, and the associated increased strain energy density. As the samples reach the fully released state, indicated as *II* in Fig. 5, the profiles for Cu-1Ta and Cu-3Ta at ~400-500 ns become narrower and steeper, approaching the ambient profile collected before testing. This indicates that the strain energy density returns to its original, unshocked state. Due to an impedance mismatch between the impactor, sample, and optical window, simulations were performed to better interpret the XRD data shown in the Supplemental File. Simulations revealed that the XRD patterns in Fig. 5B were collected for Cu-1Ta at 7.2 GPa (46% peak shock stress of 15.5 GPa) upon initial partial release from the sample-window interface, 31% and 31% peak stress during steady-state loading at 4.8 GPa, and 12 % peak stress during final release from the window free surface. For Cu-3Ta in Fig. 5B, the XRD patterns were collected at 31% and 31% peak stress during steady-state loading and 26% peak stress during full release to the unloaded state (Please refer to Supplementary Fig. 2).

In-situ time-resolved shock experiments often reveal contraction and re-emergence of the XRD profile intensity[44]. In such cases, this phenomenon is usually associated with grain rotation and realignment of grains out of the Bragg condition during the loading to the peak pressure and release to ambient conditions. However, due to the accumulated damage during these processes, the peak profile intensity does not fully return to its original state change[41]. In other words, the occurrence of stored defects in the lattice due to shock loading leads to permanent peak broadening when compared to the unshocked profile. Note that the peak broadening (the FWHM changes by as much as 24%) is accompanied by the peak shift towards higher 2θ values, indicating lattice contraction and subsequent re-expansion caused by the shock loading. As the sample undergoes unloading, the peak gradually returns to its original position. Slight peak shifting can also occur due to sample motion during the testing (shortening sample to detector distance). However, examination indicates that such small motions (measured with laser velocimetry) are significantly smaller than the observed peak shifting and can be ignored. Thus, the XRD

analysis indicates that the stabilized NC Cu-Ta microstructure can facilitate the recovery kinetics within the lattice. This testing confirms that post-mortem observations are not the result of post-testing recovery processes and, indeed, occur on the time scale of the shock and release during plate impact testing.

## Atomistic simulations of shock deformation

To gain better insights into the microstructure development mechanism during the shock loading, atomistic simulations were performed using the methodology described in the Methods section. Samples of NC Cu and Cu-3Ta alloys were equilibrated at 300 K and subjected to shock loading by the piston method. The piston velocities ranged from 0.8 – 2.0 km/s (consistent with the experiments), and the simulation times were about 100 ps to a few ns.

When the piston was moved, a shock wave was generated and propagated through the sample by compressing the atoms in the piston's direction. The stress peak magnitude was on the order of ~10–80 GPa and depended on the piston velocity and the pulse duration. The shock-induced compression caused the emission of multiple full and partial dislocations from grain boundaries and triple junctions (Fig. 6A, B). The dislocations propagated into the grain interiors and were absorbed by other grain boundaries or occasionally transferred into neighboring grains. In pure Cu, the dislocations glided freely through the grain interiors. In the Cu-3Ta alloy, the Ta clusters located at the grain boundaries and in grain interiors pinned the emitted dislocations, obstructing their nucleation and propagation and causing the vast majority of them to be absorbed back into the grain boundaries. As a result, the Cu-3Ta alloy exhibited a lower dislocation density during the compression than pure Cu. In both cases, the dislocation density increased rapidly at the shock front, then decayed after the front passed by and eventually reached a constant value (Fig. 6E–G). The residual dislocation density in Cu-3Ta was found to be significantly lower than in Cu, confirming much smaller damage accumulation.

When the shock front reached the free surface, it reflected back as a wave of tension, which eventually reached the piston and reflected back as a wave of secondary compression, and the process repeated. The tension-compression process continued with a gradually decaying stress amplitude due to the energy dissipation into heat. The magnitude of the first wave of tension is critical in determining the material failure. At the shock velocities below ~1 km/s, the shock only caused plastic deformation without fracture. After a

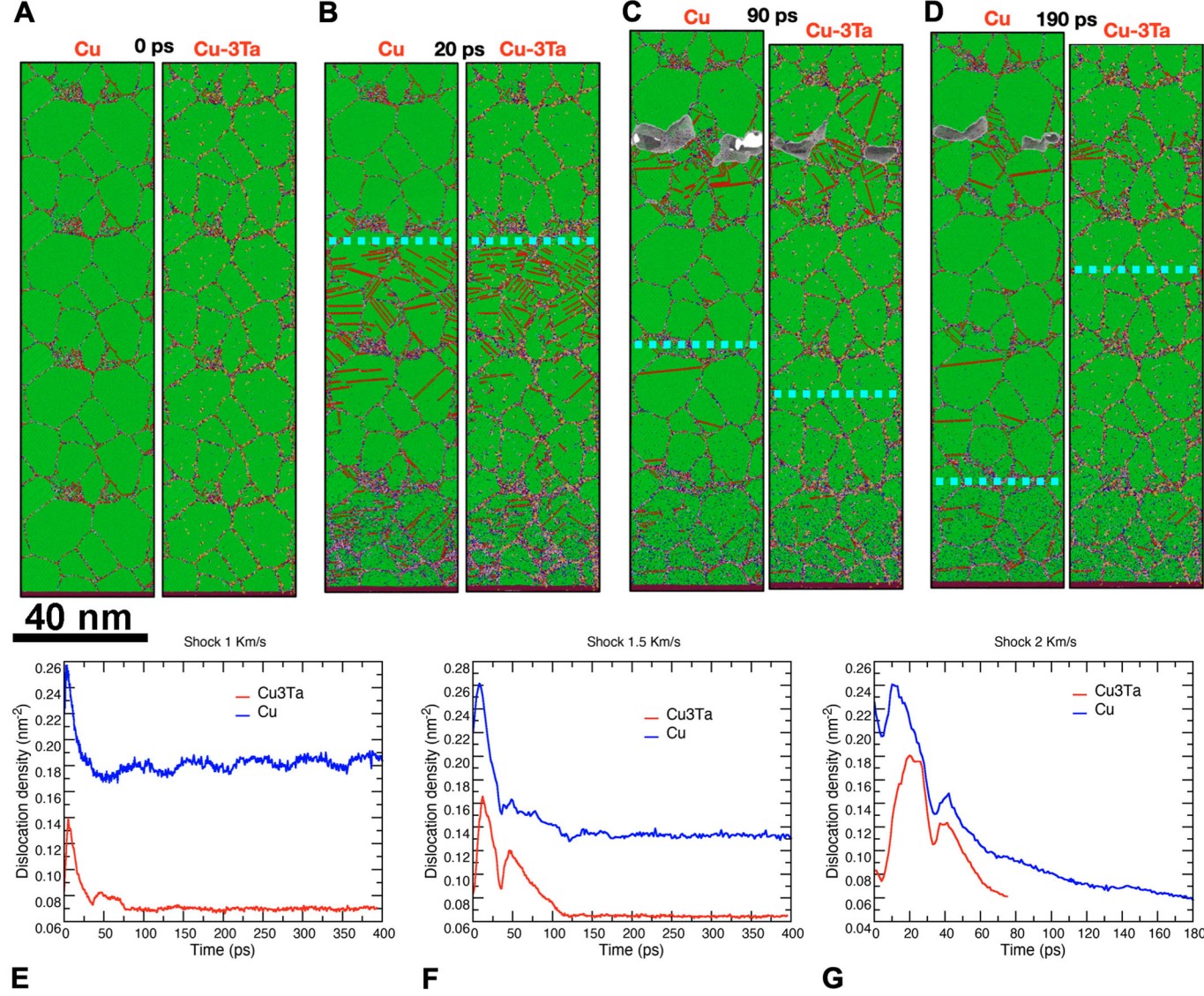

**Fig. 6 | Computer simulation of shock deformation of NC Cu and Cu-3Ta.**
**A–D** Side-by-side comparison of microstructures of Cu and Cu-Ta at different points in time. **A** Pre-shock microstructure. **B** Compression wave is in the middle of the sample. **C** Reflected tensive wave caused spallation damage. **D** After the second reflection, the voids in the Cu sample remain while the voids in Cu-3Ta have disappeared. **E–G** Dislocation density as a function of time for the shock wave velocities of E 1.0 km/s, F 1.5 km/s, and G 2.0 km/s. Note that the residual dislocation density in Cu-3Ta (red curve) is significantly lower than in Cu (blue curve). The dotted line indicates the current position of the shock front. Source data are provided as a Source Data file.

threshold value (>1 km/s), the first wave of tension caused a spallation fracture near the opposite end of the sample. The fracture started with the formation of locally disordered (amorphous) regions at grain boundaries or triple junctions that quickly evolved into a set of voids or cracks, as illustrated in Fig. 6C, D. In pure Cu, the voids remained for the rest of the simulation, even after multiple tension/compression cycles. In Cu-3Ta, the amorphous pockets developed slower and only from Ta-free grain boundary regions. The surrounding Ta clusters hindered the expansion of such pockets, delaying the spallation compared with the Cu sample. At shock velocities below ~ 2 km/s, the voids in Cu-3Ta eventually disappeared, and the material almost fully recovered from the permanent damage by re-bonding the grains along new grain boundaries and annihilating much of the high dislocation density. These findings support prior experimental results where an elevated spall strength was measured for NC Cu-3Ta compared to polycrystalline Cu[45]. At all velocities tested, the spallation started when the tensile stress reached about 10 – 12 GPa, consistent with the experimental results.

Simulated X-ray diffraction patterns collected from the microstructure were monitored during the shock tests. For example, Fig. 7 displays the time evolution of the (111) diffraction peak from a middle section of the sample (of size 40x40x40 nm³) as it was overrun by the compression wave. The profile remained unchanged until the shock wave reached the selected region. As the region came under compression, the peak broadened and shifted to the right (larger diffraction angles θ), as it should since the compression slightly decreases the lattice constant. Simultaneously, the height of the peak decreased. After the compression wave passed, the peak gradually narrowed and shifted to the left (smaller angles), approaching the initial (pre-shock) position and intensity. The process repeated as the same region was subjected to the subsequent compression waves, although the shift and broadening effects were weaker due to the reduced compression stress caused by the energy dissipation. The observed behavior of the diffraction pattern is similar to that observed in the experiments, confirming the fast dislocation relaxation processes in Cu-3Ta. The timescale in Fig. 7 is shorter than the experimental because the

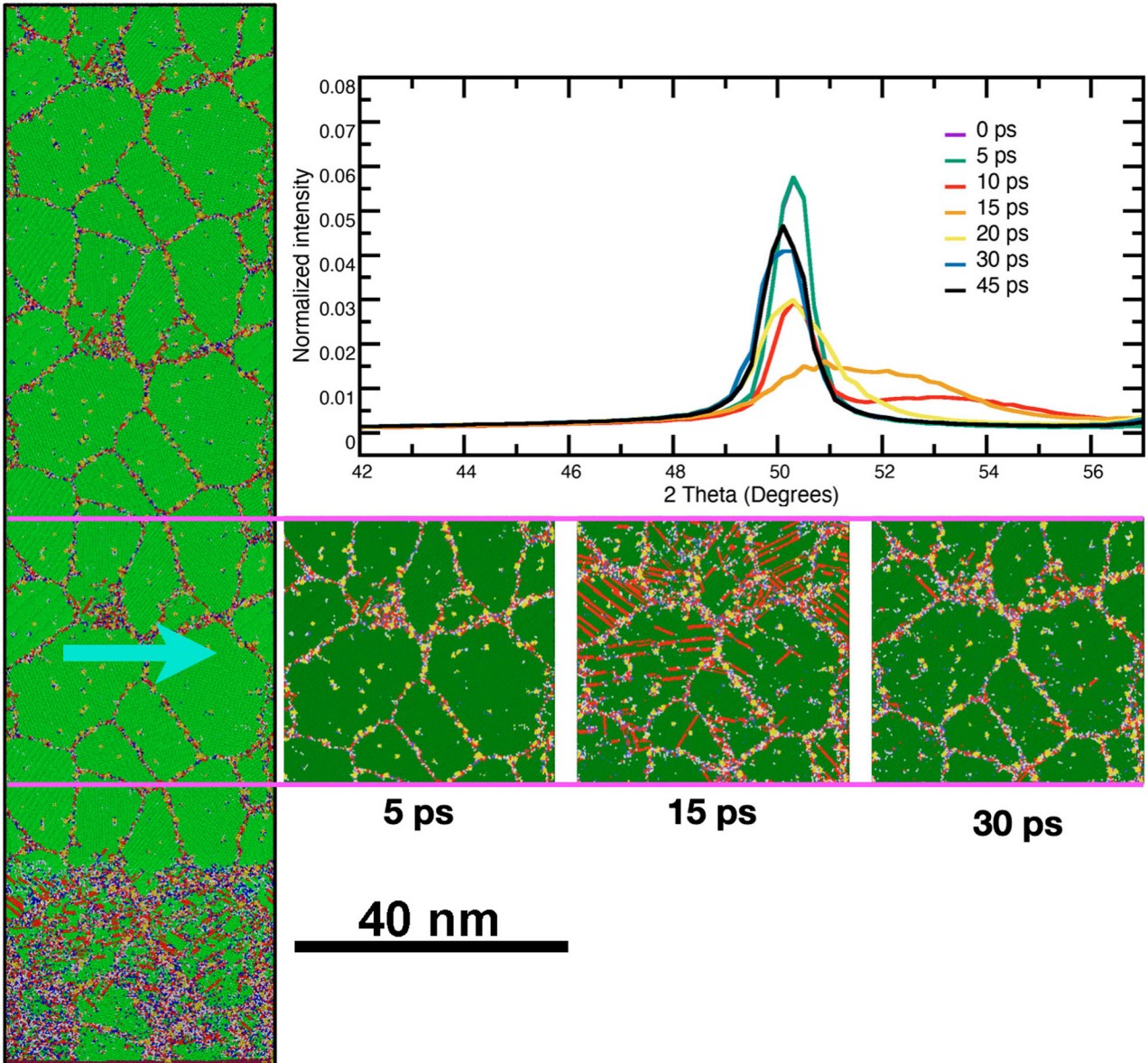

**Fig. 7 | Molecular dynamics simulation of X-ray diffraction peaks.** A middle section of the Cu-3Ta simulation sample (in the magenta frame) was chosen to follow its plastic deformation and recovery under shock loading. After the first 5 ps, the compression wave has not reached the section, and the microstructure is nearly defect-free. At 15 ps, the section is plastically deformed by the compression front. Note the numerous stacking faults created by partial dislocations. At 30 ps, the wave has passed, and the microstructure recovers the nearly defect-free state due to significant dislocation annihilation. The plot presents the time evolution of the (111) diffraction peak, showing its broadening and shift to the right followed by narrowing and return to the pre-shock diffraction angle due to plastic recovery. Source data are provided as a Source Data file.

diffraction data was collected from a narrow section of the sample. Supplementary Fig. 13 displays a similar behavior for the entire sample. This figure compared the diffraction patterns from the Cu-Ta alloy with those from pure Cu. In Cu, the initial peak is narrower but becomes broader under compression compared to Cu-Ta and returns to the original position somewhat faster (although the residual dislocation density is higher; see Fig. 6E–G). The difference is not dramatic, suggesting that the dislocation density analysis is a better method than the simulated diffraction patterns.

## Discussion

The experimental results reported in this paper demonstrate the remarkable ability of the Cu-3Ta alloy to absorb/dissipate a substantial amount of mechanical energy and damage caused by repeated shock loadings. This ability represents a previously unobserved material behavior. The capacity of NC metals to absorb energy and mitigate damage has been previously documented in studies of intense irradiation[8,46–50]. In such cases, the dense network of interfaces serves as an effective sink to capture the point defects generated during the cascading events. However, in the context of mechanical energy absorption, the practical utilization of this defect absorption ability has been hindered by the inherent instability of NC structures. NC metals are prone to grain growth during almost any form of deformation, ranging from delicate surface polishing to shock loading (Fig. 1A, B). In contrast, the NC Cu-3Ta alloy studied here is thermo-mechanically stable due to the Ta clusters pinning the boundaries and preventing grain growth. As a result, the grain boundaries continue to serve as persistent sinks for absorbing deformation-induced defects, including point defects and linear defects, such as dislocations, during the shock deformation process[24,26,46,49]. The grain stabilization is the mechanism

behind the resistance of the Cu-3Ta alloy's microstructure to damage accumulation.

Earlier atomistic studies[25,26,51] have demonstrated that in NC Cu-Ta alloys, the dislocation emission from grain boundaries and their glide into the grains during deformation are hampered by Ta nanoclusters. Many dislocations stopped by the clusters are reabsorbed by the emitting grain boundary within the simulation timeframe[11,26,51]. The present simulations further confirm this behavior and extend it to significantly higher stress levels experienced during shock loading. The Cu-3Ta alloy has displayed an increased spall strength[45], which was likewise confirmed by the present simulations. The improved spall strength is another consequence of the thermo-mechanical stabilization by the Ta clusters. It is well-known that, despite the significant advances in computational power, the space and time scales accessible by modern MD simulations are shorter than the respective experimental scales. In particular, in the present simulations, the grains were significantly smaller than in the experimental samples. As such, the dislocations emitted by grain boundaries could easily traverse the grains and be absorbed or transmitted to another grain. The probability of forming dislocation tangles inside the grains was small. In this respect, the simulations diverged from the experimental conditions, in which a spike of high dislocation density in the compressive wave could create entangled dislocations remaining in the grains longer after the shock (~100 ns time scale). This explains why the simulated dislocation structures relaxed on a much shorter time scale. Nevertheless, the simulations have captured other critical dislocation processes, especially the impact of the Ta nanoparticles on the damage resistance of the Cu-Ta alloys.

In conclusion, unlike traditional metals, thermo-mechanically stabilized nanocrystalline Cu-Ta alloys demonstrate the capacity to dissipate and exhibit either reversal or near-complete recovery from the accumulated dislocation structures, along with damage incurred, under multiple dynamic shock loading events to a high-pressure state. The microstructure of these alloys remains virtually unaffected by shock deformation. This behavior is unlikely to be displayed by other NC metals, whose microstructure loses stability under shock loading. To demonstrate the shock resistance of the Cu-Ta alloys, soft-recovered samples were subjected to repeated shock loadings at a pressure of 12 GPa. The microstructure of these alloys, especially Cu-3Ta, remained nearly damage-free after multiple shock loadings. By contrast, the UFG Cu used for comparison exhibited a drastic proliferation of defects, damage accumulation, and rapid grain growth, as observed by high-resolution STEM imaging. Time-resolved APS and XRD experiments provided direct observation of the damage recovery in Cu-3Ta after the shock loadings. Atomistic simulations corroborated the experimental results by confirming the defect generation and rapid microstructure recovery at the shock front. The simulations also linked the ability of the NC Cu-Ta microstructure to withstand severe compressions to grain stabilization by the Ta nanoclusters. The findings of this work can have significant implications for the design of new shock-resistant materials for many technological applications.

# Methods

## Powder processing and consolidation via equal channel angular extrusion (ECAE)

High-energy cryogenic mechanical alloying was used to process NC Cu-1Ta and Cu-3Ta powders. Elemental Cu and Ta powders (~325 mesh and 99.9% purity) were mixed in the appropriate ratios with milling media (440 C stainless steel balls) inside a glove box with an Ar atmosphere (oxygen and $H_2O$ levels <1 ppm). 10 g of Cu-Ta powder and appropriate amounts of media were also loaded into vials to achieve a ball-to-powder weight ratio of 5-to-1. The milling was performed using a SPEX 8000 M shaker mill at cryogenic temperature (~−196 °C) for 4 h with liquid nitrogen. The resulting NC Cu-Ta powders were consolidated to bulk using ECAE. For this process, the as-milled powder was placed in nickel cans and sealed inside the glove box under Argon atmosphere. The die assembly used for processing the billets was preheated to 623 K (350 °C) to minimize thermal loss during the ECAE process. The billets, heated and equilibrated to 973 K (700 °C) for 40 min, were quickly dropped into the ECAE tooling and extruded at a rate of 25.5 mm/s, following route $B_c$[52–54] to avoid imparting a texture to the consolidated powder. This process was repeated four times, resulting in an equivalent strain of ~4.6 and an average grain size of ~90 and 170 nm for Cu-3Ta and Cu-1Ta, respectively. In contrast, pure Cu was processed using ECAE at room temperature following route $12B_c$, resulting in an average grain size of about 350 nm after enduring an equivalent strain of ~4.6. The bar stock used for this process was oxygen-free high-conductivity (OFHC) copper with dimensions of 20 mm × 20 mm x 150 mm.

## Microstructural characterization

For the shock-compressed specimens, the site-specific lift-out technique was used, which involved a ThermoFisher Scientific Helios G4 UX dual-beam focused ion beam (FIB) / scanning electron microscope (SEM). The samples were specifically taken from the impact face for all three samples, plus the rear face and middle of the through sample thickness for the NC Cu-3Ta sample. All (S)TEM images were captured using a JEOL 2100 F microscope operated at 200 kV. All ion contrast images were also generated using the same dual-beam FIB/SEM. The images were taken with the ion column operated at 30 keV using a beam current of 26 pA and a 5 μs scan rate. For selected microstructures, STEM images were produced with ultra-high-resolution (11 k x 15 k pixels). They can be best viewed by downloading the full-resolution images available online.

## As-received microstructure

The microstructural characterization using TEM revealed nanocrystalline grain sizes for the copper and tantalum particle phases with an average grain size of 87 ± 15 nm for Cu-3Ta, see Supplementary Fig. 7. The tantalum particles/clusters exhibit a range of sizes from atomic-scale clusters ($d < 14$ nm) to larger particles ($d > 14$ nm). The larger particles and atomic-scale clusters have average diameters of 32 ± 8 nm and 3.2 ± 0.9 nm, respectively. The size distributions were obtained from areas similar to that seen in the Extended Data Fig. 2 of ref. [55] and were averaged over 300 grains. It is important to note that TEM images show all material through the thickness of the specimen such that the 3-dimensional data is projected onto a 2-dimensional image. To account for any error from such measurements, Monte Carlo simulations of the powder packing process were performed, which showed that the 2-dimensional and 3-dimensional measurements were comparable and consistent with an earlier 3D atom probe study[56]. It is noted that the microstructure has a few twins, but the formation of nano-twins during the processing was minimized due to the presence of nanoclusters[57].

## Shock sample preparation

The shock loading experiments were performed in the 80 mm light gas gun along with a three-capsule soft-recovery target fixture. The target fixture was lapped flat and mounted directly to the barrel's muzzle. A 2 mm thick OFHC copper flyer plate mounted to an 80 mm diameter sabot was accelerated to 626 m/s with compressed helium gas for the impact of the target assembly. The target and projectile were recovered in the soft-catch tank filled with rags. The velocity was calculated from recorded times of precisely measured shorting pins just before impact. Photonic Doppler Velocimetry (PDV) measurements were not performed since the fixture was designed for soft recovery, and the samples were embedded in radial and back momentum traps. However, the pulse duration obtained from simulations was ~500 ns.

The samples were confined in an OFHC copper fixture designed for soft recovery with radial and downrange momentum traps to limit

compressive and tensile loading after the initial compression wave transit. The downrange momentum trap was the same thickness as the flyer to limit the spall within the capsule. Each of the three samples was machined to right circular cylinders that fit within each of the three 25 mm capsule assemblies, Supplementary Fig. 14. The capsules consisted of a front and back disk sandwiching a ring with an inner diameter to be slip-fit with the sample. The design of the capsule assembly allowed for the use of the largest right circular sample that could be cut from the available material. The design allowed up to 10 mm diameter and 4.2 mm thick samples to be used and was adjustable for samples machined from the material recovered after the initial impact.

The peak pressure state was calculated to be ~ 12 GPa based on the standard properties of OFHC copper. The rise time of low porosity bulk metallic materials is on the order of 10 ns. The time at peak pressure is a function of the release wave transit from the up-range side of the flyer plate. For the encapsulated samples, this time was estimated to be on the order of 1 microsecond. After the first impact, the sample capsules were recovered, and the samples were removed by machining. Following post-shock characterization, the samples were re-machined into right circular cylinders and placed in the capsule assemblies for the second impact under the same impact conditions.

### Shock sample preparation for testing at the APS
Samples were thinned and polished to a mirror finish to a range of 50–57 μm. The impact experiment configuration and powder diffraction geometry used for simultaneous multi-frame powder XRD are presented in Supplementary Fig. 15. The Fig. shows that the planar shock waves in the Cu-Ta samples were generated using lithium fluoride (LiF) impactors with nominal dimensions of 4 mm thick by 10 mm diameter at velocities of at least 1.3 km/s with a 12.7 mm bore powder gun. The corresponding peak shock stresses were ~ 15 GPa. A PDV system was used concurrently to acquire the free surface velocity histories and report shock breakout time relative to the first X-ray image frame.

All in-situ XRD measurements were made using an X-ray flux spectrum peaked at 36.3 keV (0.34 Å wavelength) with a full-width-at-half-maximum bandwidth of about 0.35 keV in a single bunch of the APS storage ring operated in a 24-bunch mode. For each experiment, four indirect (phosphor-based) X-ray diffraction detectors (Active Area Diameter of 120 mm) recorded four images at an interval of 153.4 ns with an exposure time of ~100 ps to match the X-ray period in the 24-bunch mode. This time interval allows XRD images to be acquired at the four discrete times after impact. The acquired XRD data were analyzed using Dioptas software to study the microstructural evolution. All LiF spots were masked during the analysis to eliminate their effect on the integrated data.

### Atomistic simulations
The simulations employed the parallel Monte Carlo (MC) code ParaGrandMC[58] for generating the Cu-Ta microstructures and the molecular dynamics code LAMMPS[59] to perform shock deformation simulations. Atomic interactions in the Cu-Ta systems were modeled using the angular-dependent potential[60]. The visualization software OVITO[61] was used to analyze dislocations and observe the microstructure evolution during the simulations. In particular, OVITO's DXA algorithm was utilized to calculate the dislocation density. A polycrystalline copper sample composed of 32 grains with a mean grain size of 12.6 nm was constructed by the Voronoi tessellation method. The sample contained about 5.4 million atoms with approximate dimensions of $40 \times 40 \times 40$ nm$^3$ and periodic boundary conditions in all three dimensions. The grain boundary structures were optimized by the addition/removal of atoms in the boundary regions[60,62,63]. The thermodynamically equilibrium state of the Cu-3at.%Ta alloy was created by the composition-controlled MC algorithm[60,64] at a temperature

of 673 K and zero pressure. As a result, Ta atoms precipitated in the form of nanoclusters distributed predominantly at grain boundaries and triple junctions. Before the shock simulations, the sample was deformed by a 2% uniaxial tension followed by a 2% uniaxial compression. This re-treatment aimed to create more defective, and thus more realistic, grain boundary structures in the pre-shock state. This deformation also generated a small density of dislocations, mainly in the vicinity of grain boundaries. Such dislocations are barely visible simulated images shown in the paper, which only present a single cross-section of the 2D structure, but they were readily detected by the DXA algorithm and included the dislocation density plots.

For the shock simulations, the initial simulation block was replicated four times along one of the directions to create a 160 nm-long sample. After annealing the sample for 0.5 ns in the canonical (NVT) ensemble at 300 K, the ensemble was switched to microcanonical (NVE), and a shock wave was created by the piston method, treating a 1.5 nm-thick slab at one end of the simulation block as the piston. Atoms in the piston were moved as a rigid body at a fixed velocity for a time $t$ known as the pulse time. The velocity ranged from 0.8 to 2.5 km/s, and the pulse ranged from 1.5 to 20 ps $t = 1.5$. The surface boundary condition was maintained at the opposite end in the shock direction with periodic boundaries in the lateral directions. Snapshots of the simulation block were saved at regular intervals for visualization and analysis. The virtual diffraction patterns of the saved snapshots were computed using a diffraction package in LAMMPS[65] with a wavelength of 1.78901 Å, matching the experimental source Co K (alpha). The pre-shock positions and intensities of the diffraction peaks were found to be close to the experimental ones, confirming our methodology.

## Data availability
The authors declare that the data supporting this study's findings are available in the paper and its supplementary information files. Source data are provided with this paper.

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

## Acknowledgements
This work was supported by the US Army Research Laboratory under contract W911NF-15-2-0038 and the National Science Foundation under grants No. 1663287 and 1810431. Additionally, this work was performed under a cooperative agreement between the DEVCOM Army Research Laboratory and the Georgia Institute of Technology (W911NF-13-2-0018). R.K.K. and Y.M. gratefully acknowledge the financial support of the US Army Research Office, grant number ARO-W911NF-15-2-0050.

## Author contributions
B.C.H., R.K.K., G.K., P.J., N.L., J.T.L., and A.G. conducted the experiments. B.C.H., R.K.K., G.K., P.J., J.T.L., N.N.T., Y.M., and K.A.D. analyzed the data. K.S., N.N.T, and K.A.D. designed the project. B.C.H., P.J., J.T.L., K.S., N.N.T., Y.M. and K.A.D. wrote the manuscript.

## Competing interests
The authors declare no competing interests.
