## [Transparent Peer Review file · Nature Communications]

Direct observation of deformation and resistance to damage accumulation during shock loading of stabilized nanocrystalline Cu-Ta alloys

Corresponding Author: Dr Billy Hornbuckle

Version 0:

Reviewer comments:

Reviewer #1

(Remarks to the Author)

Overall the research in this paper is sound, but there are some areas that are left unclear or need further analysis. The introduction in particular is vague, and does little to set the scene of the following research or discuss damage formation under shock loading. Instead there is hyperbole concerning "advancing our society" (line 53) and inconsistencies - "advancements in materials science have opened up new frontiers" (line 69) is soon contradicted by line 82, "progress in developing such materials has been limited despite considerable efforts". Both relate to fusion materials. I would suggest that the authors rewrite the introduction. The word "remarkable" is used 7 times.

More detailed comments and questions by section:

Results – soft capture experiments

Line 141 – states soft recovery was performed at 15 GPa, but methods (line 469) gives ~12 GPa - which is it?

Line 155 "excessive grain growth" – provide some quantitative measurements to validate your finding

Line 156 "water colored grayscale" – what does this mean?

Line 161 – Cu₃Ta has a more homogenous microstructure – while Figure 1 supports this, there is a clear gradient in grain size in supplemental Figure 3 for all samples. How did grain size vary through the full 4.2mm thickness of the samples? It's been demonstrated often that stored work can be a function of distance from the impact plane

At these stresses there will be significant residual heat from the shock. Have the authors considered how this could affect grain size and dislocation content? Was there a quench system used in the soft recovery?

Results – in situ experiments

How thick were the CuXTa samples for the DCS shots? In Figure 6, there is a lineout at only 1 ns that shows full compression of the sample – is this time relative to impact or shock arrival at the sample/LiF interface? Line 284 suggests relative to impact but this should then be probing a mostly unshocked / ambient sample, which the lineout suggests otherwise

Line 288 – peak broadening discussion. The authors state there is a clear broadening and then contraction – this is not clear from the lineouts. Please quantify / measure FWHM and give an actual comparison

Figure 6 – the as-recorded diffraction patterns add no value to the paper

Discussion

The paper uses the term "heal" often. This suggests that the material undergoes a microstructural change, then somehow reverts to its original form. However, the conclusion is that the material reaches a steady state of dislocation generation and absorption, with no change to the microstructure. I don't think heal is the correct term, this would be more a resistance to damage.

Line 473 – was there PDV on the 80mm shots to verify the pulse duration?

Line 492 – there is only one scintillator / intensifier, which is then relayed to the 4 cameras via a series of beamsplitters, as in supplemental Figure 6

Reviewer #2

(Remarks to the Author)

This manuscript investigates microstructure stability of nanocrystalline Cu-Ta subjected to repetitive shock loading. NC Cu-Ta was prepared by a power-metallurgy route including cryogenic milling of the powders and consolidation via equal channel angular extrusion. These NC samples have grain sizes below 100 nm and even finer Ta clusters. The prepared Cu-1Ta and Cu-3Ta were shock loaded and soft-recovered to examine the deformation microstructure. The surprisingly high microstructure stability of the NC samples subjected to shock loading is indeed appealing and very impressive. However, there are several concerns pending to be addressed before I can recommend the publication of the manuscript.

It is shown by a series of scanning transmission electron microscopy, ion-contrast imaging and time-resolved shock experiments at APS that the microstructure is extremely stable. This is the major selling point of the paper. However, the reviewer is still confused as to what actually caused the ultrahigh microstructure stability. How come the microstructure after shock loading remains relatively unchanged? Where are the TEM samples extracted, are they adjacent to shock surface or near the rear surface? Shock wave deformation is essentially nonequilibrium and decays as a function of propagation depth. Can the authors confirm that the microstructure and mechanical behavior (hardness) is uniform everywhere across the sample?

The key observation from postmortem STEM observation seems to be the lack of defect accumulation ability for NC Cu-Ta. As to comparison of the deformed structure between Cu-3Ta and steel (Fig.5), I am not sure why the authors picked steel rather than nanocrystalline pure Cu.

The in-situ time resolved shock experiments are indeed helpful in terms of unrevealing the kinetics of the defect evolution, i.e. defects accumulation during loading and relaxation during release. The authors claim that the mechanism of damage absorption appears to take place within a few hundred nanoseconds during unloading process. What are the driving forces for the defect dissipation process? It is postulated that the interfaces such as stabilized grain boundaries and Ta clusters are the sinks for point defects and dislocations. If this is true, this works for quasistatic loading, meaning the materials would demonstrate a weak strain hardening ability. Can you confirm this?

Note that the structure stability of the same materials has been reported previously by some of the authors in the current manuscript (without detailed mechanisms) .

B. C. Hornbuckle, C. L. Williams, S. W. Dean, X. Zhou, C. Kale, S. A. Turnage, J. D. Clayton, G. B. Thompson, A. K. Giri, K. N. Solanki, K. A. Darling, *Commun. Mater.* 2020, 1, 1

Indeed current investigation is more systematic and more sophisticated experiments were carried out, but it would be beneficial to include a more detailed discussion of the energy dissipation mechanism in this manuscript. For a high profile journal such as nature communications, this should be a baseline.

Reviewer #3

(Remarks to the Author)

Please see attached review.

Version 1:

Reviewer comments:

Reviewer #2

(Remarks to the Author)

The reviewer went through the revised manuscript, all the questions raised by the reviewer (and two other reviewers) has been addressed properly. As a result, the manuscript has been modified accordingly and the quality of the manuscript markedly improved. Specifically, the mechanism of the deformation and energy dissipation, which was not well presented in the previous version, is illustrated reasonably by the MD simulation. Therefore, the reviewer is convinced that it can be published by the Nature Communications.

Reviewer #3

(Remarks to the Author)

This resubmission contains some excellent modifications which address many of the reviewer's concerns. However, there are still significant issues with the interpretation of the dynamic in-situ experiments and the newly added atomic simulations. I don't think the proposed mechanism surrounding the "healing" is well supported.

1. The addition of the Sinclair et al. (2021) data in Figure 5a could be extremely misleading and I don't think it is an appropriate comparison. This experiment was in a different XRD setup (obvious from the 2Theta values) and a different experimental configuration consisting of LiF impactor, 152 micron thick copper, and a sapphire window. The peak stresses and temporal evolution of the stress history is significantly different from the NC data in B and C. The peak stress in the Sinclair data is higher (19 vs 15.5 GPa) and release state will be at a significantly higher stress due to the impedance difference between sapphire and PMMA. This is not an apples-to-apples comparison. To provide conclusive evidence

OFHC Cu is behaving differently from the Cu-1Ta and Cu-3Ta systems, the same experiment should be performed where the only difference is the sample material.

2. I'm not convinced the authors have explained all the complexities of the in-situ shock XRD patterns. In the rebuttal the authors state "Furthermore, while the majority of the issues listed by the Reviewer have to be considered for the APS experiments, the beamline is highly a monochromatic beam which helps to greatly minimize the issues commonly associated with XRD line broadening experiments." This is not true. As explained in Sinclair et al. "For XRD experiments at the DCS, the undulator is typically configured to deliver a high X-ray flux, 7×10^{15} photons s⁻¹, at 23 keV. Since no monochromator is used to filter this beam, the high peak flux during a bunch precludes the use of photon-counting area detectors." DCS experiments use a pink beam and I reiterate my previous comment: interpreting line broadening and peak intensity in these experiments is extremely difficult due to the non-monochromatic nature of the beam coupled to the dynamic effects on the Bragg condition.

3. My biggest criticism is that the story the authors are trying to tell does not make sense. The proposed mechanism is the dislocations generated to accommodate the plasticity are reabsorbed quickly behind the shock front. The implication from the MD simulations in Fig 7 is this should happen on the 10s of ps timescale. If this were the case then shouldn't all of the measured patterns in Fig. 5 look like the ambient curve? It is only after the sample returns to zero pressure (~ 500 ns) that the claimed recovery happens. I also reiterate my point about the Sinclair et al. OFHC Cu data: without knowing the stress history (and when it drops to 0) this is not a valid comparison.

4. I thank the authors for generating the requested figures in Supplementary Fig. 2, but this highlights my previous point: the spatial and temporal evolution of the stress within the sample is complicated and the majority of the XRD measurements are examining the state of the sample after it has reverberated to the partial release state of 4.8 GPa. This is not consistent with probing a region of constant stress behind the shock wave like the MD simulations. Moreover, if the claim is the $t < 400$ ns patterns in Fig.5 are demonstrating damage in the material, then it has yet to "heal" despite timescales orders of magnitude larger than the atomic simulations.

5. The new atomic simulations are not consistent with mechanism being proposed. The results in Fig. 6 demonstrate the overall dislocation generation is lower in the Cu3Ta scenarios but the qualitative nature of the deformation is the same. Both exhibit a rapid increase in dislocation density at the shock front followed by a similar decay to a steady value. The results are on a linear scale and so the relative difference (a factor of 2-3) really isn't that much as it pertains to the dislocation mechanics. The mechanism proposed is a complete "healing" or reabsorption of the dislocations so shouldn't the dislocation density be dropping back to zero to support the hypothesis? I also find the discussion of tension to be completely irrelevant and distracting as it does not pertain to either the recovery or in-situ shock experiments.

6. Fig 6 and Fig 7 seem like they have some inconsistencies. Again, Fig. 6 does not show a complete dislocation annihilation which is the claim in Fig. 7. Is there another mechanism that could be resulting in the evolution of the simulated diffraction pattern? Perhaps due to dynamic crystallographic orientation/texture changes? Also, I think it is extremely important to also show the corresponding simulated diffraction for the pure Cu simulation. Does it show a significantly different behavior like the hypothesis would suggest?

Version 2:

Reviewer comments:

Reviewer #3

(Remarks to the Author)

My previous review centered around several ideas presented by the authors and how they are supported by the dynamic diffraction measurements and molecular dynamics simulations. In this revision, I do not believe my concerns were addressed and I stand by my assertions that several of the key claims (highlighted below) are not well supported.

In the first sentence motivating the In-situ shock experiments section, the authors state, "A series of shock loading experiments were carried out at the Dynamic Compression Sector of Argonne National Laboratories' APS [44] to confirm that the absence of residual dislocations in the Cu-Ta samples was not due to post-shock reversal of bowed dislocations." My interpretation of this sentence along with the rest of the article is the authors are claiming there is an absence of residual dislocations directly behind the shock front. In other words, this is not due to a reversal in the loading/release in the pressure behind the shock front. I contend the DCS data are not sufficient to quantitatively extract the appropriate information to argue one way or another, while the MD simulations are in direct conflict with the author's conclusions.

1. I maintain the analysis here is not nearly detailed enough to disentangle the many effects that could contribute to the measured diffraction patterns, especially defect evolution.

a. In their rebuttal the authors cite 4 references they claim study line broadening at DCS. I strongly disagree that any of these studies provide an example of using line broadening at DCS to infer microstructural deformation characteristics. In references 1 (10.1103/PhysRevX.10.011010) and 2 (10.1103/PhysRevX.10.011010) the metric is lattice spacing (i.e. diffraction peak locations) not broadening. Moreover, both works forward-model the pink beam to interpret the results and the latter uses Rietveld refinement. This is the level of analysis I understand to be necessary to interpret these types of experiments and is why I'm skeptical of the claims the authors are making. I further highlight a statement in reference 2:

"Note that the diffraction patterns observed in the current study exhibit characteristic peak broadening compared to the simulated patterns (dotted patterns), due to the width and asymmetry of the pink x-ray (non-monochromatic) beam". The authors here are not considering these types of complexities and aren't even using quantitative metrics, only that they observed some broadening and some narrowing of the diffraction profiles. I also note the experimental data never fully releases to zero pressure (and such a release would still be at elevated temperature), so I don't understand how comparisons to the ambient pattern are at all meaningful.

b. Reference 3 (10.1557/jmr.2015.65) is not DCS and reference 4 (10.1103/PhysRevB.104.064113) is a wonderful article detailing the complexities of understanding a XRD pattern taken at DCS. The excerpt the authors quote highlight some of these complexities, but this article does not discuss how to actually account for them. Again, this is my whole point – the authors aren't performing a rigorous analysis considering all the potential complexities so how do they know they are making the correct interpretation?

c. I think there is an argument to made that if XRD patterns were collected at similar times (i.e. loading histories) for both a Cu-Ta and pure Cu system and measurable quantitative differences were made for the broadening then perhaps something could be inferred about the difference in defect accumulation between the two materials. However, I think the authors would agree this would be extremely difficult to diagnose if the MD simulations are to be believed as in the last paragraph of the rebuttal they discuss the difference between the simulated diffraction patterns between pure Cu and the Cu-Ta alloys: "The difference is not dramatic, suggesting that the dislocation density analysis is a better method to evaluate the difference than the simulated diffraction patterns".

d. Finally, even if one accepts the author's interpretation of the diffraction profiles is correct, I strongly disagree with what is being stated. The patterns are being collected during pressure release (46-12% of peak stress) and so the measurements are not representative of what is happening directly behind the shock front. In other words, I think this is evidence that the recovery of the dislocation structure could be directly correlated to the post-shock reversal of bowed dislocations. This is exactly opposite of the hypothesis the authors present.

2. I better understand what is happening with the MD simulations but it still isn't clear if the results of interest are happening within the steady high pressure region of the pulse or in the region where the pressure drops to zero (i.e. what is the pulse duration, t , for the results presented?). This gets at the fundamental question I've tried to relay in my reviews and I still don't have clarity on: is the mechanism of interest here active directly behind the shock front (i.e. in the steady Hugoniot state) or is it the result of the later time dynamic drop to low pressure? My take on Fig. 7 is the 30ps defect free microstructure is occurring after the pulse duration time and is therefore at zero pressure. Again, this is directly contrary to the hypothesis the authors are presenting which is "the absence of residual dislocations in the Cu-Ta samples was not due to post-shock reversal of bowed dislocations." These types of contradictions are why I've been so adamant that the authors are not clearly supporting their conclusions.

a. I greatly appreciate the discussion on the MD length and timescales and feel the authors covered this very well. However, I don't understand what the significant differences are between the Cu and Cu₃Ta MD results. To my eye, the results in Fig. 6 look qualitatively the same, particularly the defect structures post-shock and post-release. Isn't the pure Cu simulations also showing recovery of the initial defect structure and thus a similar mechanism? There is a quantitative offset in the dislocation density (~2x) that the authors say confirm the smaller damage accumulation, but isn't this offset roughly uniform in both the shock and release regimes? A dashed horizontal line on Fig.6 e-g showing the initial dislocation density may be extremely helpful to distinguish differences between the starting and late-time dislocation densities between the two materials. Fig. 13 suggests any differences in the diffraction pattern are very small - arguably within experimental uncertainties? Given all of this, I still don't understand how the MD results portray a dramatic change in the underlying mechanism or the novel phenomenon this article is centered around. Don't get me wrong, the mechanism is clearly there in reality as proven in the recovery aspects of the work, but I don't think it is being demonstrated definitively in these MD results. Perhaps that is the comment related to length and time scales, but my understanding is this effect should have only been exaggerated by the small MD grain sizes?

b. I appreciate the MD results pertaining to damage under tension are an intriguing aspect of these simulations, but they are completely removed from the rest of the article. I disagree with the argument that this article is about "the material's resistance to diverse forms of damage accumulation". This article is a multi-faceted but focused study on uniaxial compressive deformation (which is a compliment and a strength of the article) with an MD tension result thrown in at the end. Given the caveats with the MD, this result adds little while adding a good amount of complexity and a mindset switch to follow. Again, my recommendation is to move this out of the main text and to a different article or to supplementary information.

Version 3:

Reviewer comments:

Reviewer #4

(Remarks to the Author)

I have read and reviewed the manuscript titled "Direct observation of deformation and resistance to damage accumulation during shock loading of stabilized nanocrystalline Cu-Ta alloys". The manuscript is well-written, comprehensive, and logical with well-justified conclusions. In the opinion of this reviewer, the manuscript should be accepted as is (presumably with higher resolution figures).

Direct Observation of Deformation and Damage Healing during Shock Loading of Stabilized Nanocrystalline Cu-Ta Alloys

Manuscript NCOMMS-23-30586A

Response to Reviewers

We are grateful to the reviewers for the thorough and insightful comments and suggestions. We have rewritten the Introduction section and changed the title to remove the vagueness and better focus on the results while simultaneously toning down the hyperbole. We have also addressed all comments and questions by the Reviewers and revised the manuscript accordingly. The revisions required additional analyses of the experimental data, which are included in the revised manuscript. We have also performed atomistic simulations supporting the experimental study and included the results in the revised manuscript. Additionally, we have performed finite element simulations to extract the pressure and velocity-time history of the sample relative to x-ray exposure times for the Advance Photon Source (APS) experiments. Since the simulations were used primarily to characterize the transient nature and order-of-magnitude estimates of velocities and pressures, existing Steinberg-Guinan equations of state were utilized. As a result, the revised manuscript and the Supplementary Information File contain a significant amount of new text and several new figures. With the additional simulation work, three new authors have been added to the author list.

Reviewer #1

(1) Results - soft capture experiments Line 141 - states soft recovery was performed at 15 GPa, but methods (line 469) gives ~12 GPa - which is it?

We apologize for the typo, which resulted in the inconsistency between the reported peak pressure values. The correct value is 12 GPa and has been corrected in the Methods section of the revised manuscript.

(2) Line 155 “excessive grain growth” - provide some quantitative measurements to validate your finding

The average grain size in the pre-shocked materials was 350 nm for OFHC Cu. The post-shocked soft-recovered specimen of OFHC Cu had an average grain size of ~20 μm , which is a ~57 times increase in the average grain size. This speaks to the instability of OFHC Cu's grains to deformation-induced grain growth as opposed to that observed in Cu-1Ta and Cu-3Ta alloys. These metrics have been added to the revised manuscript for clarity. Electron Backscattered Diffraction (EBSD) was utilized to generate the grain orientation maps to determine the average grain size for the impact surface, middle, and rear surface of the specimen. These maps and the grain distribution plots are included in the Supplementary Figure 6.

(3) Line 156 “water colored grayscale” - what does this mean?

We use this terminology, water-colored grayscale, to describe the variation in contrast within the coarsened OFHC Cu grains resulting from Focused Ion Beam Channeling Contrast Imaging (FIB-CCI). The contrast in the image is a result of variations in channeling efficiency, which is related to the orientation and perfection of the underlying crystal lattice. In the case of shocked soft-recovered specimens, certain grains maintain a consistent, either light or dark contrast across their entire surface. However, other grains exhibit subtle shifts in contrast as one transverses them due to fluctuations in stored dislocation density. These localized strain accumulations induce alterations in the local crystal orientation and lattice parameters, consequently manifesting as visual differences in contrast. Here to avoid confusion, we have replaced “water colored grayscale” with “variation in contrast.”

(4) Line 161 – Cu₃Ta has a more homogenous microstructure – while Figure 1 supports this, there is a clear gradient in grain size in supplemental Figure 3 for all samples. How did grain size vary through the full 4.2mm thickness of the samples? It's been demonstrated often that stored work can be a function of distance from the impact plane at these stresses there will be significant residual heat from the shock. Have the authors considered how this could affect grain size and dislocation content? Was there a quench system used in the soft recovery?

Indeed, we only reported the microstructural evolution at the impact face. We believe the gradient effect is caused by intense frictional forces resulting from the interaction of the

impact plate with the front face of the sample. Additionally, these forces also occur between the rear face of the sample and the backing plate during shock loading (refer to Supplementary Fig. 14). We thank the reviewer for pointing this out and giving us the opportunity to provide additional characterization of the rear surface (opposite to the impact surface) as well as the midpoint of the through-thickness. These additional images of the grain size have been included in the Supplemental Information File (Supplementary Fig. 6, 8, and 10). They confirm the initial results and trends for the 3 different materials at various locations away from the impact surface. As can be seen in the middle of the sample, no gradient in the grain size exists.

The reviewer is correct that the stresses experienced during the shock can produce a temperature rise. No quench system was used in the soft recovery of the samples. In the shock recovery experiments, the computed temperature rise is 57°C for a duration of approximately 600 nanoseconds. In the x-ray diffraction experiments, the peak computed temperature rise is 76°C, which transiently fluctuates over 30 nanoseconds as the waves bounce back and forth from the PMMA window, reaching a steady temperature rise of 41°C. The sample is held at this temperature for approximately 400 nanoseconds until the release wave from the back face of the PMMA window arrives, further reducing the temperature. Based on the computed temperature rise and duration, it does not appear that residual heat from the shock should significantly alter the microstructure between the time it is arrested and the time that the microstructure is measured experimentally. Furthermore, such time is not sufficient to allow for the diffusion-based relaxation processes, as typically observed with annealing out of dislocations at high temperatures.

(5) Results - in situ experiments How thick were the CuXTa samples for the DCS shots? In Figure 6, there is a lineout at only 1ns that shows full compression of the sample - is this time relative to impact or shock arrival at the sample/LiF interface? Line 284 suggests relative to impact but this should then be probing a mostly unshocked / ambient sample, which the lineout suggests otherwise

The thickness for the different samples tested at the DCS ranged from 50-57 μm . The 5 ns lineout was a typo and has been corrected. The times listed are with respect to (computed time of impact) based on the time of shock breakout from PDV at the rear sample surface and wave transit time. That noted data frame was captured 17 ns after shock breakout, or 29 ns after impact. Based on velocimetry as measured at the rear

surface, the peak pressure is reached within 12-13 ns. Although the early time point does not represent the material state at peak pressure, the shift in peak position to higher 2θ is consistent with compression early in shock loading.

(6) Line 288 - peak broadening discussion. The authors state there is a clear broadening and then contraction - this is not clear from the lineouts. Please quantify / measure FWHM and give an actual comparison.

For clarity, the measured change in FWHM is now provided in the manuscript to be more quantitative. However, we would like to state that since the area under any given diffraction peak is conserved, a change in the overall intensity of the diffraction peak must also result in a change in the FWHM. Thus, a change in peak height is proportional to a change in FWHM. As can be seen, there is a large apparent change in the peak intensity present in Figure 5.

As stress and strain are accrued, lattice planes of individual grains may be distorted or contain defects. These distortions and defects create local strain fields within the grains, causing variations in the lattice spacing. When X-rays interact with these distorted regions, peak broadening occurs due to the non-uniform lattice spacing within the grains. Also, increasing XRD peak intensity- (the decreasing of FWHM)- means higher levels of ordering and structural enhancement.

(7) Figure 6 - the as-recorded diffraction patterns add no value to the paper

We have removed the diffraction images (upper row of panels) but kept the peak profiles. We want the reader to see the actual profiles extracted from the measurements. The behavior observed in the XRD patterns for NC CuTa alloys deviates from the texture evolution observed in shocked Cu.

(8) Discussion: The paper uses the term “heal” often. This suggests that the material undergoes a microstructural change, then somehow reverts to its original form. However, the conclusion is that the material reaches a steady state of dislocation generation and absorption, with no change to the microstructure. I don’t think heal is the correct term, this would be more a resistance to damage.

We regret any prior confusion and have made efforts to provide a clarification. The mechanism discussed does not involve simultaneous generation and absorption of defects in a steady state. Instead, it entails the generation of dislocations, followed by a subsequent process of reabsorption over a period of time after the shock induced deformation. The dislocations nucleated, glide some distance into the matrix, are arrested by the Ta cluster, and then finally, at later times, reabsorbed by the nucleating boundary. The new simulations presented in the manuscript illuminate the specific mechanisms underlying this delayed absorption phenomenon in the Cu-Ta alloys, which stands in contrast to pure Cu and its inability to undergo these processes.

(9) Line 473 - was there PDV on the 80mm shots to verify the pulse duration?

No, PDV measurements were not performed in these experiments since the fixture was designed for soft recovery, and the samples were embedded in radial and back momentum traps. However, the pulse duration obtained from simulations is ~500 ns.

(10) Line 492 - there is only one scintillator / intensifier, which is then relayed to the 4 cameras via a series of beamsplitters, as in supplemental Figure 6

Yes, the four-frame detector system includes a single scintillator imaged by 4-cameras using beam splitters. As detailed by Sinclair et al. (2021), the front end of the system is made up of a fast X-ray phosphor-coated fiber optic plate, fiber optic taper, and image intensifier. The back end of the system is made up of beam splitters, four camera lenses, and four ICCD cameras. Additional information on the camera system and technique can be found in Sinclair et al. *Journal of Synchrotron Radiation* (2021) <https://doi.org/10.1107/S1600577521003775>. We have also included an updated, i.e., slightly modified description of the system in Supplementary Figures 14 and 15.

Reviewer #2 (Remarks to the Author):

This manuscript investigates microstructure stability of nanocrystalline Cu-Ta subjected to repetitive shock loading. NC Cu-Ta was prepared by a powder-metallurgy route including cryogenic milling of the powders and consolidation via equal channel angular extrusion. These NC samples have grain sizes below 100 nm and even finer Ta clusters. The prepared Cu-1Ta and Cu-3Ta were shock loaded and soft-recovered to examine the

deformation microstructure. The surprisingly high microstructure stability of the NC samples subjected to shock loading is indeed appealing and very impressive. However, there are several concerns pending to be addressed before I can recommend the publication of the manuscript.

(1) It is shown by a series of scanning transmission electron microscopy, ion-contrast imaging and time-resolved shock experiments at APS that the microstructure is extremely stable. This is the major selling point of the paper. However, the reviewer is still confused as to what actually caused the ultrahigh microstructure stability. How come the microstructure after shock loading remains relatively unchanged? Where are the TEM samples extracted, are they adjacent to shock surface or near the rear surface? Shock wave deformation is essentially nonequilibrium and decays as a function of propagation depth. Can the authors confirm that the microstructure and mechanical behavior (hardness) is uniform everywhere across the sample?

We apologize for not explaining more clearly the reason for the microstructural stability. The reason for the microstructure remaining unchanged is the ability of the grain boundaries to generate dislocations and then reabsorb them during the shock loading. The grain boundaries are strongly pinned (via Zener pinning) by the sub-10 nm Ta clusters. The proof of the Ta clusters' role can be seen from the substantial microstructural changes in the Cu sample versus the minor changes observed in the microstructure of Cu-1Ta alloy and no discernible microstructural change in the Cu-3Ta alloy due to its optimized amount of Ta. These aspects have been made clearer in the revised manuscript.

The TEM samples were taken directly from the impacted surface of all 3 specimens. We have performed additional TEM and FIB-CCI imaging of the rear face as well as halfway through the thickness of the specimen. In addition to the microstructural imaging, we have performed additional hardness measurements across the thickness of the sample.

These images of the grain size and hardness profile have been added to the Supplementary Information File, and these confirm the results and trends for the 3 different materials at various locations away from the impacted surface.

(2) The key observation from postmortem STEM observation seems to be the lack of defect accumulation ability for NC Cu-Ta. As to comparison of the deformed structure

between Cu-3Ta and steel (Fig.5), I am not sure why the authors picked steel rather than nanocrystalline pure Cu.

In Figures 1-3, we compared the microstructure of UFG Cu with that of NC Cu-Ta alloys before and after shocking to highlight the difference in the grain size stability between the two. These figures show active grain growth in pure Cu, indicating that Cu no longer retained its initial microstructure and exhibited defect accumulation. Our rationale was that comparing NC Cu-Ta alloy's deformation structures to that of steel, which is viewed as a more resilient material than pure Cu due to not undergoing grain growth, would further emphasize the lack of microstructural changes in NC Cu-Ta. However, we decided not to pursue the comparison with steel in the revised manuscript. Thus, Figure 5 has been removed.

(3) The in-situ time resolved shock experiments are indeed helpful in terms of unrevealing the kinetics of the defect evolution, i.e. defects accumulation during loading and relaxation during release. The authors claim that the mechanism of damage absorption appears to take place within a few hundred nanoseconds during unloading process. What are the driving forces for the defect dissipation process? It is postulated that the interfaces such as stabilized grain boundaries and Ta clusters are the sinks for point defects and dislocations. If this is true, this works for quasistatic loading, meaning the materials would demonstrate a weak strain hardening ability. Can you confirm this?

To better address the reviewer's question about the driving forces for the defect dissipation, we have now included in the revised manuscript atomistic simulations (using molecular dynamics (MD) and Monte Carlo). The simulations provide more insight into the mechanisms operating during the loading and release of the shock pressure. The simulations show the generation of dislocations in the NC Cu-3Ta microstructure as the shock wave passes through followed by their absorption back into the grain boundaries. The specific mechanisms and driving forces are also discussed in the manuscript.

We also appreciate the astute question related to quasistatic loading and weak strain hardening ability. Indeed, NC Cu-Ta alloys have been shown to exhibit a near elastic perfectly plastic response under quasi-static loading exhibiting minimal, if any, strain

hardening as reported in (K.A. Darling et al. / Acta Materialia 76 (2014) 168–185 as well as other previous work).

(4) Note that the structure stability of the same materials has been reported previously by some of the authors in the current manuscript (without detailed mechanisms. B. C. Hornbuckle, C. L. Williams, S. W. Dean, X. Zhou, C. Kale, S. A. Turnage, J. D. Clayton, G. B. Thompson, A. K. Giri, K. N. Solanki, K. A. Darling, Commun. Mater. 2020, 1, 1. Indeed current investigation is more systematic and more sophisticated experiments were carried out, but it would be beneficial to include a more detailed discussion of the energy dissipation mechanism in this manuscript. For a high profile journal such as nature communications, this should be a baseline.

We appreciate the critique and are glad the reviewer has given us a chance to elaborate and incorporate more discussion in the manuscript, specifically related to the energy dissipation mechanism. This discussion focuses on NC Cu-Ta alloy's ability to generate dislocations when experiencing the shock pressure and then quickly dissipate this energy/deformation by reabsorbing the dislocations. A more in-depth discussion became possible with the addition of the atomistic simulation work in the revised manuscript. This additional work corroborated the APS results while providing insight into the mechanisms at play within the material during the energy dissipation process. Please see the revised manuscript for changes.

Reviewer #3 (Remarks to the Author):

Review of “Direct Observation of Extreme Mechanical Energy Transfer and Healing of Damage from Repeat Shock-Loading in Stabilized Nanocrystalline CuTa Alloys”

This article describes an interesting set of results on a nanocrystalline Cu-Ta alloy that suggests it can be shock compressed to 15 GPa at least 2 times without a significant change to the microstructure. As the authors note, this observation is unique within the landscape of how other metals behave and this work may offer insights into future applications where materials can be tailored to withstand multiple shock loading events. While the authors show some compelling results and I believe the implications to be

sufficiently broad and impactful to consider for publication in Nature Communications, I have several questions and comments which resulted in my recommendation of significant revisions (particularly item #5):

(1) Within the realm of possible explanations of the results, the authors do not discuss why purely elastic deformation not plausible. The authors inherently assume the deformation is plastic and is mediated by dislocations, but I do not understand the justification for this. I recognize the authors have published a wide range of properties for variants of this material, but to my knowledge the Hugoniot elastic limit remains unknown and there isn't definitive evidence of plastic flow within the Cu-3Ta experiments. References 9 and 28 provide information on flow stress as a function of deformation rate for similar materials, but they do not extend to the shock regime. If there is a mechanism change due to the high deformation rates associated with shock loading and dislocation mediated plasticity is not the primary deformation mechanism, then this dramatically changes the interpretation of the work; I think it is crucial that the authors rule this out.

The reviewer raises a fair point of whether the Cu-3Ta has undergone plastic deformation with the current data presented. We acknowledge an oversight on our part in not including an additional reference (2022 Acta paper on Spall) where the authors determine the HEL and spall strength of Cu-3Ta to be 2.0 GPa and 1.19-1.67 GPa, respectively. Also, we have included the time-resolved particle velocity profile generated from the APS data using Photonic Doppler Velocimetry (PDV). This data confirms the deformation to be plastic. Additionally, while there were no PDV measurements made on the 80 mm shots, calculations were performed, and the pressure profiles generated at different locations in the impacted target yielded peak shock pressures of 12 GPa. The shock pressure is much higher than the HEL (2 GPa), where you would expect plasticity. See Supplementary Fig. 2 for more detail.

(2) a. On a related note: I think it is worth mentioning some ceramics as a class of materials that exhibit elastic deformation over the 15 GPa shock stresses of interest in this article. I believe a similar set of experiments on a high strength ceramic would be expected to have a similar outcome where multiple shock loadings would result in little change to the initial microstructure, but the reason is the deformation remains elastic. I suggest this

not to diminish the impact of the results here, but to enhance them by highlighting the difference from materials typically used to solve similar engineering challenges. Metals offer many unique advantages and a better understanding of the physics governing the “healing” observed in these experiments could open the possibility for these metals to vastly outperform their ceramic counterparts.

We thank the reviewer for this insightful note regarding similar behavior in ceramic materials due to 15 GPa shock remaining within these materials’ elastic regimes. We agree with the reviewer’s opinion. This comparison enhances the results due to harnessing the superior shock resistance properties of ceramics, while simultaneously maintaining the ductility associated with metals despite undergoing a high pressure shock loading condition (12 GPa) without detrimentally experiencing shock hardening generally seen in metals during plastic deformation via shock. We have also included new MD simulations, which provide a better understanding of the actual physics governing the observed healing.

(3) It is stated many times, such as in the abstract, “The microstructural features of nanocrystalline Cu-Ta are shown to withstand repeated shock-wave interactions at pressures up to 15 GPa without any evidence of microstructural damage or deterioration,...”. Figures 2 and 3, however, contain red circles showing regions of damage and dislocation activity for the Cu-1Ta and Cu-3Ta alloys in both single and second shock scenarios. Isn’t this evidence of observed deterioration? Perhaps the wording is just too strong, and these are measurable but small differences?

The reviewer makes valid points regarding our wording choice when discussing microstructural damage or deterioration of Cu-1Ta and Cu-3Ta alloys. We have toned down the wording to state that the slightest amount of dislocations indicative of plastic deformation are present within Cu-3Ta alloy, while Cu-1Ta alloy does have dislocations present as well as regions of abnormal grain growth. However, when comparing these microstructures with those of post-shocked pure Cu, they exhibit negligible change despite also undergoing plastic deformation.

(4) I really appreciate the general approach in this article of showing results for pure NC copper, Cu-1Ta, and Cu-3Ta as it highlights the broad expectations of how conventional

NC metals behave in contrast to the Ta alloys. Table 1 was an odd change from this pace. Why are the ECAE OFHC Cu and NC-Cu-1Ta results not included?

We thank the reviewer for the kind words regarding the approach of comparing NC Cu-Ta alloy to conventional NCs and the vast difference in their respective behaviors. However, given that these two microstructures have undergone some measurable microstructural evolution (e.g., grain growth and increased dislocation accumulation), the hardness trends are more difficult to interpret. Few metals and alloys (e.g. Ni-based) display softening following shock compression. NC metals, however, exhibit extreme hardness relative to their coarse-grained counterparts, e.g. 10-20X higher. This is solely due to their much finer grain structure. For example, conventional coarse grain Cu will exhibit a Vicker hardness/Yield stress of 60 MPa, while nanocrystalline Cu is ~ 800 MPa. However, as shown in this work, these types of microstructures are extremely unstable under shock loading. The ensuing loss in yield strength/hardness due to the drastic coarsening (two orders of magnitude increase in grain size) can be much larger than that gained by storing dislocations from shock hardening. Therefore, it makes comparing ECAE OFHC Cu and NC-Cu-1Ta to examples reported in the literature of shock-hardened material not practical. Since NC Cu-3Ta alloy's grain size is maintained, one can directly compare its residual hardness change to other metals, as in Table 1.

(5) a. Also related to the VHN measurements, why did the uncertainty for the NC-CU-3Ta result change from 291 +/- 4.84 VHN to 304 +/- 9.73 VHN between the pre and post shocked material? I bring this up because the increased uncertainty brings these into reasonable agreement, while an uncertainty of 5 in both scenarios puts this closer to a measurable difference.

The reviewer has a very keen eye for noting this difference in increased uncertainty for the post-shocked material. This likely resulted in some confusion. The largest uncertainty (± 9.73) is plus or minus 3.2% of the average value. The lowest uncertainty (± 4.84) is plus or minus 1.6% of the average value. Buehler/Wilson, the manufacturer of the indenter, provides a certificate of calibration with the instrument, certifying the instrument's error to be 1.6% of the average.

The reported hardness values in this study were obtained from the as-impacted surface, which tends to be slightly rougher after shock testing. As pointed out by Reviewer #1, there seems to be a minor gradient in grain size very close to the free surface, varying in both continuity and depth, spanning from 0.5 to 1.5 microns. It's worth noting that this gradient appears to become more pronounced after the second shock, as seen in Supplementary Figures 5 and 6 for comparison. The variability in surface roughness, as well as this gradient, likely contributes to the slight increase in the measurement's uncertainty.

Originally, the samples were left unpolished to preserve them for potential future shock testing. However, in response to the reviewer's feedback, the authors decided to cut, mount, and polish the samples for further hardness testing. The newly presented data exhibits better agreement with the pre-shocked hardness values for Cu-3Ta alloy with lower uncertainty. Please note that the microstructural images taken from the middle of the sample thickness do not show a microstructural gradient, which again could affect the initially reported hardness value taken from the impact surface.

(6) I don't fully understand the reasoning behind including Figure 5 and the surrounding discussion. I understand the point about how pervasive crystallographic defects are observed in shock compression of conventional metals (and the broad VHN results in Table 1 are a succinct way to make this point), but I think the pure Cu results are the most impactful here. Steel is a completely different system with its own complications, so I think it has minimal impact, may cause confusion, and does not warrant space in the primary text.

Other Reviewers have also questioned the inclusion of the steel micrograph with well-justified reasons in line with the one above. We have removed the reference to steel and thus Fig. 5 since it was distracting.

(7) The biggest weakness in this paper is the APS experiments described in Figure 6. The work leading up to this detailing the work on the shock recovered samples (STEM and diffraction) is well-done and creates a clear picture that the microstructure in the recovered samples does not change significantly when comparable to that of the starting

material. I have several comments and concerns about the dynamic in-situ measurements, though:

a. The objective of these experiments is to (line 278) "...ensure that the lack of preserved dislocation activity in the Cu-3Ta, as revealed by microscopy and XRD analysis of soft recovered samples, is not due to reversal effects following shock compression...". My interpretation of this statement along with the proposed mechanism accommodating the deformation is that dislocations are emitted behind the shock front to accommodate the plasticity but are then quickly pinned by the Ta clusters and reabsorbed by the grain boundaries – resulting in the "healing" of the metal. What is not clear from the APS data or the discussion is what the role of the load reversal is (ie. the process that takes the sample back down to zero pressure). Is the release wave the driving force for the "healing" or is it proposed the metal is "healed" directly behind the shock front. I believe this is an important distinction and at the heart of what these experiments are intended to probe but the results are ambiguous in this regard.

The Reviewer is spot on with their question/critique about the mechanism responsible for the 'healing' within the metal. To provide better insight into this topic, we have included atomistic simulations of NC Cu-Ta undergoing shock loading within a similar time frame as the APS experiments (Supplementary Fig.13). The simulations indicate that the shock front generates multiple dislocations to accommodate the plasticity. Once the front passes by, the majority of the dislocations are reabsorbed.

(8) A big part of the problem is it isn't clear what the temporal evolution of the stress history in the sample looks like. Line 286 states "...compared to the time resolved XRD data obtained during both the loading and unloading states (as specified in the plot legend)", but the plot legend only refers to times and it isn't clear when the sample is experiencing the steady stress associated with the supported shock, when it is unloading from the release from the flyer/sabot interface, and when it reaches the steady release pressure. I'd recommend leveraging the 2-D simulations alluded to earlier in the article to quantify the stress states of the region of sample corresponding to the x-ray beam path as a function of time to better communicate what is actually being measured.

We would like to thank the reviewer for this assessment. We provide additional information below, in addition to modifications of the manuscript. Supplementary Fig. 2 show X-t plots for a 1d simulation of the pressure and velocity history within the sample and in the surrounding setup during the relevant course of deformation for the diffraction experiments. As the reviewer can see, the sample first experiences a transient shock period as waves reverberate between the LiF/Cu/PMMA interfaces. The peak shock of approximately 15.5 GPa is experienced for only a small duration of 30 ns, with subsequent equilibration to a steady-state pressure of 4.8 GPa for approximately 400 ns. Unloading from the back surface of the PMMA window occurs at approximately 440 ns, prior to any lateral release waves emanating from the outer diameter of the sample, which would arrive around 700-1000ns. The text and supplementary figures have been modified accordingly to point the reader more accurately to the stress history that the sample has experienced for specific times that diffraction images are obtained. Additionally, we have added the MD simulations with corresponding discussion describing the microstructural evolution as a function of time relative to that seen in the APS profiles.

(9) Related to the points above, is the release state thought to be an important feature in what is being measured? This seems to contrast with the primary objective quoted in line 278 as part of comment (a) in the sense that the XRD profiles of interest are all at late times where the material appears to be reversing. Moreover, it isn't even clear what the 1-D release pressure is. What I mean by this is I believe the experiments are designed to exist in a state of uniaxial strain during the times measured in the APS experiment, but the experimental configuration suggests the steady release pressure is related to the flyer/sabot impedance matching (ie. it is not a free surface release and so it is not zero). I don't think the authors have ruled out the effects from the edge release waves (subsequent ringing/damping) and that truly bring the sample down to zero pressure where subsequent STEM and XRD analysis was performed.

The delayed shifting of peaks back to the ambient position is due to the experimental configuration and associated impedance-mismatch, which leads to non-zero stress release until almost 500 ns, i.e. the sample is released from peak pressure but held above the HEL for 300 - 400 ns. The experimental configuration was chosen to ensure sufficient X-ray counts, i.e. 50 um thick specimen, and precluded the use of a more X-ray absorbing optical window such as sapphire. However, the primary feature of note is that following

plastic deformation, gains in peak intensity are observed as the peaks shift to their ambient positions upon full release, which suggests that the lattices are free of damage/deformation. This is in stark contrast to what is observed in OFHC Cu in prior APS work by Sinclair et al. (2021). The steady-state pressure was approximately 4.8 GPa based on shock simulations and does not fully release until after 400 ns. It is important to note that the shock recovery testing and subsequent STEM/XRD analysis are separate from the APS experiments, where only in-situ data can be collected. Both the in-situ data and post-mortem recovery XRD, as well as post-mortem STEM, reveal lattices free of damage following plastic deformation. The lateral sample dimensions are approximately 9 mm, so lateral release occurs much later than that seen in the data collected in the APS experiments.

(10) Finally, it isn't clear how the APS experimental results are actually informing this work. What is observed to my admittedly naive eye is a decrease in the peak magnitude and shift to right at early times followed by a partial recovery of the ambient peak at later times. This is consistent with a compression of the lattice and then return to its initial lattice spacing, but I struggle to understand how anything else can be extracted.

We feel that the context of the APS experiments is now better highlighted by the addition of the new MD simulations. The APS experiments confirm what is observed in the post-mortem analysis and confirm that the observed results are not due to post-shock recovery mechanisms. They also lay the groundwork for future work where shock pressures will be explored that render sample recovery intractable. The fact that the peaks are observed to shift and decrease in height with peak broadening indicates deformation and damage. The peaks are observed to both shift back to the ambient position and regain their original intensity, indicative of a stress-free lattice. This time-resolved microstructural evolution is not observed by in-situ XRD of OFHC pure Cu, and likewise, it is not possible to observe the effects during post-mortem analysis.

(11) The authors suggest there is peak broadening from 1 to ~300ns but this isn't obvious or quantified. Moreover, it isn't clear that any broadening must be the result dislocation nucleation. How are these results supporting the proposed mechanism here? My

understanding is these types of XRD experiments are extremely difficult and line broadening and peak intensity are notoriously challenging to fully model due to issues like the dynamic effects on the Bragg condition (like the authors mention) or the non-monochromatic nature of the x-ray beam used here.

Please see the response to Reviewer #1, question (6)

Furthermore, while the majority of the issues listed by the Reviewer have to be considered for the APS experiments, the beamline is highly a monochromatic beam which helps to greatly minimize the issues commonly associated with XRD line broadening experiments. The addition of the pressure time profiles, as well as the time-resolved free surface velocity profiles, indicate the samples are plastically deformed. Since the last time frame shows the fundamental peak is nearly identical to that of the pre-shocked condition, we can assume that no dislocations were stored in the microstructure despite their being generated to support plastic deformation. We would like to point out that in all other cases of shock-loaded metals, extensive non-equilibrium concentrations of defect are found stored in post-shock recovered samples.

(12) Why would the fully released state return to the ambient peak? Again, the experimental configuration suggests this does not release to zero pressure, but some finite value associated with the flyer/sabot properties.

The experiment is only capable of capturing 4 or 5 frames at predetermined time intervals during the actual shock experiments. Therefore, we attempted to capture the XRD scans as close to the shock event itself, while also trying to capture the profile of the material as great a time as possible after the shock and as close as possible to zero pressure within the experimental confines available to them. Furthermore, the newly incorporated 1D and 2D simulations in the Supplementary Information File offer insights into the time required for the system to return to near-zero pressure.

(13) The peak magnitudes never fully recover like in the static scenario (Fig. 4). So, how do these results support the conclusions in the paper that these in-situ measurements reveal the “healing” of the material on the hundreds of ns timescale?

While the reviewer is correct, the beamline experiments do not show full recovery. However, this is why we included the conventional XRD spectrum for the material pre and post- (1st and 2nd) shocked, which are fully recovered due to reaching zero pressure. The purpose of the APS experiment was to try and capture the time evolution of the absorption of dislocations which now is better informed by the MD simulations. We hope the new simulations and additional supporting information give more credence to the APS data.

Response to Reviewers

“Direct observation of deformation and resistance to damage accumulation during shock loading of stabilized nanocrystalline Cu-Ta alloys”

Manuscript NCOMMS-23-30586A

May 13, 2024

We are grateful to Reviewer #3 for carefully reading the revised manuscript and providing additional insightful comments. The paper has been substantially revised to address the Reviewer's questions and comments. This document summarizes our responses to the Reviewer and the changes made in the manuscript. The Reviewers' comments are reproduced verbatim and colored in blue followed by answers in black. The changes in the manuscript are highlighted in yellow.

Reviewer #3

This resubmission contains some excellent modifications which address many of the reviewer's concerns. However, there are still significant issues with the interpretation of the dynamic in-situ experiments and the newly added atomic simulations. I don't think the proposed mechanism surrounding the "healing" is well supported.

1. The addition of the Sinclair et al. (2021) data in Figure 5a could be extremely misleading and I don't think it is an appropriate comparison. This experiment was in a different XRD setup (obvious from the 2Theta values) and a different experimental configuration consisting of LiF impactor, 152 micron thick copper, and a sapphire window. The peak stresses and temporal evolution of the stress history is significantly different from the NC data in B and C. The peak stress in the Sinclair data is higher (19 vs 15.5 GPa) and release state will be at a significantly higher stress due to the impedance difference between sapphire and PMMA. This is not an apples-to-apples comparison. To provide conclusive evidence OFHC Cu is behaving differently from the Cu-1Ta and Cu-3Ta systems, the same experiment should be performed where the only difference is the sample material.

To avoid confusion in the interpretation of the APS Data, the Sinclair data has been removed. The intent of the APS testing was to confirm that post-mortem observations of defect density were not conflated by other factors related to the post-testing recovery processes. Our results indicate that the peak returns to the ambient condition without evidence of microstructural evolution upon the release back to the ambient state. This has been one of the important contributions due to facilities such as DCS, where

it was confirmed that material changes occurred over the testing time and eliminated misinterpretation of post-mortem microscopy.

2. I'm not convinced the authors have explained all the complexities of the in-situ shock XRD patterns. In the rebuttal the authors state "Furthermore, while the majority of the issues listed by the Reviewer have to be considered for the APS experiments, the beamline is highly a monochromatic beam which helps to greatly minimize the issues commonly associated with XRD line broadening experiments." This is not true. As explained in Sinclair et al. "For XRD experiments at the DCS, the undulator is typically configured to deliver a high X-ray flux, 7×10^{15} photons s^{-1} , at 23 keV. Since no monochromator is used to filter this beam, the high peak flux during a bunch precludes the use of photon-counting area detectors." DCS experiments use a pink beam and I reiterate my previous comment: interpreting line broadening and peak intensity in these experiments is extremely difficult due to the non-monochromatic nature of the beam coupled to the dynamic effects on the Bragg condition.

We appreciate the Reviewer's careful examination of the complexities of in situ diffraction at DCS. It is true that the beamline utilizes a pink beam configuration during XRD experiments, and it is expected that diffraction patterns observed in the current study exhibit a characteristic peak broadening compared to the simulated patterns due to the width and asymmetry of the pink x-ray (non-monochromatic) beam. However, XRD studies to date have utilized the current system to examine peak positions and shapes from materials with crystalline order or to demonstrate the lack of such order, with in situ measurements being used to infer dislocation activity in shocked solids through changes in diffraction patterns. Ignoring peak broadening prior to full release, differences in peak broadening are interpreted as due to differences in defects, which is confirmed by post-mortem observations of significant dislocation activity completely absent in the TEM analysis of the Cu-Ta samples. Below are examples of studies using line *broadening at DCS with in situ synchrotron radiation to provide evidence of the following deformation characteristics:*

- (1) Stacking faults - 10.1103/PhysRevX.10.011010
- (2) Phase transformation - 10.1038/s41598-022-21332-y
- (3) Defect clusters - 10.1557/jmr.2015.65
- (4) Analyzing in situ XRD 10.1103/PhysRevB.104.064113

Excerpt from (4) "...the increased broadening observed in the measured Bragg peaks—is likely a fundamental attribute of shock compression of solids. Beyond the Hugoniot elastic limit, shock compression causes inelastic deformation resulting in significant microstructural changes; the generation of dislocations [34], stacking faults [28,35], twinning [36], and other material-specific defects would cause a reduction in the dimensions of the coherently diffracting domains leading to an inherent broadening of the peaks."

3. My biggest criticism is that the story the authors are trying to tell does not make sense. The proposed mechanism is the dislocations generated to accommodate the plasticity are reabsorbed quickly behind the shock front. The implication from the MD simulations in Fig 7 is this should happen on the 10s of ps timescale. If this were the case then shouldn't all of the measured patterns in Fig. 5 look like the ambient curve? It is only after the sample returns to zero pressure ($t \sim 500$ ns) that the claimed recovery happens. I also reiterate my point about the Sinclair et al. OFHC Cu data: without knowing the stress history (and when it drops to 0) this is not a valid comparison.

This is a good question. It is similar to questions 4 and 5 below, which also point to the time scale disparity between the experiment and the MD simulations. Please see our responses to those questions below.

Also, as indicated above, we have removed the data from the paper by Sinclair et al.

4. I thank the authors for generating the requested figures in Supplementary Fig. 2, but this highlights my previous point: the spatial and temporal evolution of the stress within the sample is complicated and the majority of the XRD measurements are examining the state of the sample after it has reverberated to the partial release state of 4.8 GPa. This is not consistent with probing a region of constant stress behind the shock wave like the MD simulations. Moreover, if the claim is the $t < 400$ ns patterns in Fig. 5 are demonstrating damage in the material, then it has yet to “heal” despite timescales orders of magnitude larger than the atomic simulations.

The MD simulations also capture several reverberations of the wave, like in the experiment. However, we agree with the Reviewer that there is a significant disparity in the time scales of the damage recovery processes in the experiments and MD. The cause of this disparity and how we interpret the MD simulations despite the time scale disparity is discussed in our answer to the next question and in the new text added to the revised manuscript.

5. The new atomic simulations are not consistent with mechanism being proposed. The results in Fig. 6 demonstrate the overall dislocation generation is lower in the Cu₃Ta scenarios but the qualitative nature of the deformation is the same. Both exhibit a rapid increase in dislocation density at the shock front followed by a similar decay to a steady value. The results are on a linear scale and so the relative difference (a factor of 2-3) really isn't that much as it pertains to the dislocation mechanics. The mechanism proposed is a complete “healing” or reabsorption of the dislocations so shouldn't the dislocation density be dropping back to zero to support the hypothesis? I also find the discussion of tension to be completely irrelevant and distracting as it does not pertain to either the recovery or in-situ shock experiments.

There are three questions here, which we will answer in order.

Question 1. *Dislocation mechanisms of post-shock recovery.* To our understanding, three processes occur during the shock compression:

(a) Dislocations (full or partial) bow out from the grain boundaries, extending into the grains. This process can be partially reversible if the dislocations retract back to the boundaries. While relatively easy in pure Cu, this process is hampered in the Cu-Ta alloys by the Ta clusters acting as obstacles to the dislocations gliding into the grains.

(b) Dislocation emission by grain boundaries and their glide across the grains with absorption by other boundaries or transfer into a neighboring grain. This process is largely irreversible and is, again, significantly suppressed by the Ta clusters acting as obstacles.

(c) Dislocation entanglement inside the grains resulting from impingement of dislocations mentioned in (b). Such tangles are only partially reversible and contribute to the residual dislocation density left after the shock.

The MD simulations reported in the paper qualitatively confirm the mechanisms (a) and (b), and especially the role of the Ta clusters. As such, these simulations play a significant complementary role in the interpretation of the experimental data. However, due to the well-known length and time scale limitations of MD, the grains simulated here are too small to accommodate the entangled dislocations. This can explain the underestimated peak dislocation density under the strong compression and the faster-than-experimental timescale of the dislocation density drop after the shock.

This discussion has been added to the paper on page 17 (top paragraph) for the sake of clarity.

Question 2. *The complete "healing" term.* We did not claim that the damage healing was *complete* in either simulations or experiments. The damage healing demonstrated in this work was only partial (including the void healing observed in the simulations). However, to avoid any confusion, we have removed the term "healing" both in the title of the paper and throughout the text. Instead, we use more specific terminology, pointing to the more significant reduction in the dislocation density after the shock of the Cu-Ta alloys compared with pure Cu. As discussed below (Reviewer's question 6), the dislocation density in the post-shock samples does not drop to exactly zero in either experiments or simulations.

Question 3. *Damage under tension.* The in-situ shock experiments are indeed a significant part of this paper. However, our message is more general. It highlights the material's resistance to diverse forms of damage accumulation. We thus believe that the discussion of spall damage under tension is relevant, and we would prefer to keep it in the paper.

6. Fig 6 and Fig 7 seem like they have some inconsistencies. Again, Fig. 6 does not show a complete dislocation annihilation which is the claim in Fig. 7. Is there another mechanism that could be resulting in the evolution of the simulated diffraction pattern? Perhaps due to dynamic crystallographic orientation/texture changes? Also, I think it is extremely important to also show the corresponding simulated diffraction for the pure Cu simulation. Does it show a significantly different behavior like the hypothesis would suggest?

We have not claimed a *complete* dislocation annihilation in Fig.7 or elsewhere. However, we see the Reviewer's point and have rephrased the description of the damage recovery to avoid any confusion. We now use descriptions such as "annihilating much of the high dislocation density" and "significant dislocation annihilation."

Fig. 6 shows that the pure Cu sample is characterized by a higher initial dislocation density, a higher peak density, and a high residual dislocation density after the shock wave has passed a few times. It may indeed seem surprising that the sample contains a nonzero dislocation density prior to the shock deformation, as shown in the graphs in Fig. 6 (e f, g), while such dislocations are not visible in Fig. 6(a) and 7. The reason is that we routinely pre-strain our simulated samples before the shock tests (in fact, before any mechanical tests) to create a more realistic structure. Without the pre-straining, the grain boundary structures are unrealistically too perfect, resulting in a large stress overshoot and similar artifacts during the subsequent deformation. We should have mentioned this step when describing the

sample preparation. We have now added this information in the Methods section. The pre-straining produces a small dislocation density, which is not easily visible in images such as Fig.6(a) and Fig.7 showing a single cross-section of the 3D structure. However, this dislocation density is automatically detected by the DXA algorithm and is plotted in Figures 6(e,f,g). For the same reason, the residual dislocation density is not visible in Fig.7 (30 ps panel), but Figures 6(e,f,g) show that it exists, although it is much lower than the peak density.

We agree with the reviewer that the simulated diffraction patterns should be shown for pure Cu for comparison with the Cu-Ta alloys. Such patterns have been calculated and included in the revised manuscript in Supplementary Fig. 13. In Cu, the initial peak is narrower but becomes broader under compression in comparison with Cu-Ta and returns to the original position somewhat faster (although the residual dislocation density is higher, see Fig. 6 (e,f,g)). The difference is not dramatic, suggesting that the dislocation density analysis is a better method to evaluate the difference than the simulated diffraction patterns. These comments are added to the top paragraph on page 16.

Response to Reviewers

“Direct observation of deformation and resistance to damage accumulation during shock loading of stabilized nanocrystalline Cu-Ta alloys”

Manuscript NCOMMS-23-30586C-Z

June 26, 2024

My co-authors and I are grateful for the time and effort Reviewer #3 put into this most recent review. However, we do not agree with Reviewer #3's review. His or her comments are solely focused on the APS experiments and the corresponding MD simulations. With that said, even if we were to completely remove those results from the paper, the conclusion of the paper would remain unchanged based on the conventional shock, XRD, and microstructural characterization provided in the manuscript. This document summarizes our responses to Reviewer #3 and the basis for our disagreement with his or her review. The Reviewers' comments are reproduced verbatim and colored in blue followed by answers in black.

Reviewer #3 (Remarks to the Author):

My previous review centered around several ideas presented by the authors and how they are supported by the dynamic diffraction measurements and molecular dynamics simulations. In this revision, I do not believe my concerns were addressed and I stand by my assertions that several of the key claims (highlighted below) are not well supported.

In the first sentence motivating the In-situ shock experiments section, the authors state, “A series of shock loading experiments were carried out at the Dynamic Compression Sector of Argonne National Laboratories' APS [44] to confirm that the absence of residual dislocations in the Cu-Ta samples was not due to post-shock reversal of bowed dislocations.” My interpretation of this sentence along with the rest of the article is the authors are claiming there is an absence of residual dislocations directly behind the shock front. In other words, this is not due to a reversal in the loading/release in the pressure behind the shock front. I contend the DCS data are not sufficient to quantitatively extract the appropriate information to argue one way or another, while the MD simulations are in direct conflict with the author's conclusions.

1. I maintain the analysis here is not nearly detailed enough to disentangle the many effects that could contribute to the measured diffraction patterns, especially defect evolution.

a. In their rebuttal the authors cite 4 references they claim study line broadening at DCS. I strongly disagree that any of these studies provide an example of using line broadening at DCS to infer microstructural deformation characteristics. In references 1 (10.1103/PhysRevX.10.011010) and 2

(10.1103/PhysRevX.10.011010) the metric is lattice spacing (i.e. diffraction peak locations) not broadening. Moreover, both works forward-model the pink beam to interpret the results and the latter uses Rietveld refinement. This is the level of analysis I understand to be necessary to interpret these types of experiments and is why I'm skeptical of the claims the authors are making. I further highlight a statement in reference 2: "Note that the diffraction patterns observed in the current study exhibit characteristic peak broadening compared to the simulated patterns (dotted patterns), due to the width and asymmetry of the pink x-ray (non-monochromatic) beam". The authors here are not considering these types of complexities and aren't even using quantitative metrics, only that they observed some broadening and some narrowing of the diffraction profiles. I also note the experimental data never fully releases to zero pressure (and such a release would still be at elevated temperature), so I don't understand how comparisons to the ambient pattern are at all meaningful.

b. Reference 3 (10.1557/jmr.2015.65) is not DCS and reference 4 (10.1103/PhysRevB.104.064113) is a wonderful article detailing the complexities of understanding a XRD pattern taken at DCS. The excerpt the authors quote highlight some of these complexities, but this article does not discuss how to actually account for them. Again, this is my whole point – the authors aren't performing a rigorous analysis considering all the potential complexities so how do they know they are making the correct interpretation?

c. I think there is an argument to be made that if XRD patterns were collected at similar times (i.e. loading histories) for both a Cu-Ta and pure Cu system and measurable quantitative differences were made for the broadening then perhaps something could be inferred about the difference in defect accumulation between the two materials. However, I think the authors would agree this would be extremely difficult to diagnose if the MD simulations are to be believed as in the last paragraph of the rebuttal they discuss the difference between the simulated diffraction patterns between pure Cu and the Cu-Ta alloys: "The difference is not dramatic, suggesting that the dislocation density analysis is a better method to evaluate the difference than the simulated diffraction patterns".

d. Finally, even if one accepts the author's interpretation of the diffraction profiles is correct, I strongly disagree with what is being stated. The patterns are being collected during pressure release (46-12% of peak stress) and so the measurements are not representative of what is happening directly behind the shock front. In other words, I think this is evidence that the recovery of the dislocation structure could be directly correlated to the post-shock reversal of bowed dislocations. This is exactly opposite of the hypothesis the authors present.

Authors' Response

Part 1 - Reviewer #3 endlessly argues the APS data is not convincing of what is occurring to the generation and/or absorption of the dislocations and its effect on the generated diffraction patterns. The authors are using the APS data to further inform the prior microstructural characterization presented in the manuscript. This data clearly delineates a change in the dislocation density and grain size for the microstructurally unstable OFHC Cu, partially stabilized Cu-1Ta, and fully stabilized Cu-3Ta. These respective specimens were all shock recovered simultaneously to ensure no variation in their shock stress history after two iterative rounds of

shock experiments. While the authors would love the ability to conduct an in-situ shock experiment in the TEM, this is technology not currently available for use. Consequently, the APS work concentrated on collecting the in-situ diffraction patterns for Cu-1Ta and Cu-3Ta since their microstructure remains stable.

Part 1a. - This goes directly to Reviewer #3's comment on 'disentangle the many effects that could contribute to the measured diffraction patterns, especially defect evolution'. Since OFHC Cu also has microstructural changes during shock loading, this complicates interpreting the broadening it experiences compared to Cu-1Ta and Cu-3Ta, which is only due to the compression of its lattice during the shock loading. The shift of their peaks to higher 2θ positions during the in-situ is completely based on the lattice of the Cu-1Ta and Cu-3Ta undergoing compression. While the reviewer notes the peak broadening due to the width and asymmetry of the pink X-ray (nonmonochromatic) beam, which obviously is a concern for the APS data, this is not the case for the conventional XRD profiles for Cu-1Ta and Cu-3Ta pre and post-2nd shock. These diffraction patterns clearly show the broadening and decrease in peak intensity of Cu-1Ta due to retained dislocations within the grains that grew. Decoupling of these two events is difficult since they are intertwined with each other. Yes, the reviewer is correct. The samples were never fully released (it is not as if the author intentionally did not show that aspect, but rather, this is a limitation of the beamline they have access to), but again, the shock-recovered TEM samples are from the fully released condition. Thus, one must look at the entire dataset, not as individual pieces, to stand alone by themselves. Regarding the not fully released sample also being at higher temperatures, the authors have done extensive annealing studies of the Cu-3Ta (1000 hours at 800C 0.8T_M) with no microstructural changes occurring to the alloy to further ease the reviewer's concerns on that front.

Part 1b. - It is a bit insulting that the reviewer states: 'Again, this is my whole point – the authors aren't performing a rigorous analysis considering all the potential complexities so how do they know they making the correct interpretation?' On top of the data presented here, the authors of this manuscript have published over 20 other peer-reviewed articles on various aspects of nanocrystalline Cu-Ta. Specifically, 3 others have been published regarding the shock behavior of Cu-3Ta. Thus, the authors do not take lightly the responsibility of publishing a result on questionable or unsubstantiated data. Rather, they strive to have substantial results to base their conclusions on. Thus, the authors respond to this absurd reviewer's comment by questioning their understanding of nanocrystalline materials and their response to extreme stimuli, which is apparent from their numerous comments during the review process.

Part 1c. – The reviewer states: 'I think there is an argument to made that if XRD patterns were collected at similar time (i.e. loading histories) for both a Cu-Ta and pure Cu system and measurable quantitative differences were made for broadening then perhaps something could be inferred about the difference in defect accumulation between the two materials. However, I think the authors would agree this would be extremely difficult to diagnose if the MD simulations are to be believed as in the last paragraph of the rebuttal they discuss the difference between the simulated diffraction patters between pure Cu and the Cu-Ta alloys: "The difference is not dramatic, suggesting that the dislocation density analysis is a better method to evaluate the difference than the simulated diffraction patterns".

The authors can only shake their heads at these comments for several reasons. First, the shock recovery experiments and the microstructural characterization of the pure Cu, Cu-1Ta, and Cu-3Ta reported in the first half of the manuscript are the exact experiments that the reviewer is requesting. In those reported microstructural results between pure and Cu-3Ta, there is a **DRAMATIC** difference in not only dislocation density but also grain size. The micron-size grains present in the fully recovered pure Cu are due to deformation-induced grain growth, which is a highly recognized issue within nanocrystalline materials and the primary reason for their lack of incorporation within the industry. The authors included the MD simulations to better explain the APS data and appease the reviewer on top of the already abundant evidence from the microstructural characterization of what occurred within Cu-3Ta. The MD results confirm the significantly reduced dislocation density in the shock-recovered Cu-3Ta relative to that in the pure Cu sample, in agreement with the experimental data. The MD also provides further details on the dislocation processes during and immediately after the shock deformation. Finally, the authors do not have a beamline accessible to perform the requested in-situ experiment; however, if the reviewer has access or wants to build such a beamline, the authors would be happy to take them up on the opportunity to perform such an experiment.

Part 1d. – The authors respect the reviewer’s opinion but could not disagree more with this comment. The percentage of the peak stress measured for the samples again is not the authors’ attempt to hide data or mislead the reviewer, but rather the simple limitations of the APS experiment. The authors included this data as further evidence of CuTa’s ability to generate and then reabsorb most of the dislocations during shock experiments, as seen by the post-microstructural characterization presented throughout the manuscript and supplemental section, not as the only experimental evidence for the conclusions drawn by the authors. This is the greatest frustration the authors have with the reviewer’s comments. The APS data plays a complementary role relative to the microstructural characterizations and MD simulations. **Even if the APS data were withdrawn from the manuscript, this would not change the main conclusions of this work.**

Reviewer #3 (Remarks to the Author):

2. I better understand what is happening with the MD simulations but it still isn’t clear if the results of interest are happening within the steady high pressure region of the pulse or in the region where the pressure drops to zero (i.e. what is the pulse duration, t , for the results presented?). This gets at the fundamental question I’ve tried to relay in my reviews and I still don’t have clarity on: is the mechanism of interest here active directly behind the shock front (i.e. in the steady Hugoniot state) or is it the result of the later time dynamic drop to low pressure? My take on Fig. 7 is the 30ps defect free microstructure is occurring after the pulse duration time and is therefore at zero pressure. Again, this is directly contrary to the hypothesis the authors are presenting which is “the absence of residual dislocations in the Cu-Ta samples was not due to post-shock reversal of bowed dislocations.” These types of contradictions are why I’ve been so adamant that the authors are not clearly supporting their conclusions.

a. I greatly appreciate the discussion on the MD length and timescales and feel the authors covered this very well. However, I don’t understand what the significant differences are between the Cu and

Cu₃Ta MD results. To my eye, the results in Fig. 6 look qualitatively the same, particularly the defect structures post-shock and post-release. Isn't the pure Cu simulations also showing recovery of the initial defect structure and thus a similar mechanism? There is a quantitative offset in the dislocation density (~2x) that the authors say confirm the smaller damage accumulation, but isn't this offset roughly uniform in both the shock and release regimes? A dashed horizontal line on Fig.6 e-g showing the initial dislocation density may be extremely helpful to distinguish differences between the starting and late-time dislocation densities between the two materials. Fig. 13 suggests any differences in the diffraction pattern are very small - arguably within experimental uncertainties? Given all of this, I still don't understand how the MD results portray a dramatic change in the underlying mechanism or the novel phenomenon this article is centered around. Don't get me wrong, the mechanism is clearly there in reality as proven in the recovery aspects of the work, but I don't think it is being demonstrated definitively in these MD results. Perhaps that is the comment related to length and time scales, but my understanding is this effect should have only been exaggerated by the small MD grain sizes?

b. I appreciate the MD results pertaining to damage under tension are an intriguing aspect of these simulations, but they are completely removed from the rest of the article. I disagree with the argument that this article is about "the material's resistance to diverse forms of damage accumulation". This article is a multi-faceted but focused study on uniaxial compressive deformation (which is a compliment and a strength of the article) with an MD tension result thrown in at the end. Given the caveats with the MD, this result adds little while adding a good amount of complexity and a mindset switch to follow. Again, my recommendation is to move this out of the main text and to a different article or to supplementary information.

Authors' Response

2. This long and winding paragraph shows that the reviewer is utterly confused about the capabilities and interpretation of MD simulations. The authors are at a loss since every additional step they take to answer the reviewer's comments/concerns just leads to more questions and concerns despite the reviewer literally stating, 'as proven in the recovery aspects of the work.' Most of these additional questions are already answered in the paper.

For example, as indicated in the text, the MD pulse duration was a few picoseconds. The 15 ps image in Fig. 7 represents a Hugoniot state, while the 30 ps image represents the post-shock state. The Hugoniot state is characterized by a high density of partial bow-out dislocations as evidenced by the high density of the intrinsic stacking faults (red stripes). At 30 ps, most of the partial dislocations are gone, although probably not all because the image only shows a single cross-section of the microstructure. (The same behavior is seen in Fig.6(a-d) comparing Cu-Ta and Cu after much longer simulations.) More importantly, the image visualizes the grain boundaries and the Ta nanoclusters and is blind to full dislocations. This is why we applied the DXA analysis to reveal all dislocations and quantify their density, with the results presented in Fig.6(e-g). The results show a drastic decrease in the dislocation density after the shock recovery and the residual dislocation density that is much smaller in Cu-3Ta than in Cu. Most of the bowed dislocations present in the Hugoniot state have disappeared. The plots cover the time scales of hundreds of ps, orders of magnitude longer than the pulse duration. This explanation almost literally repeats what

is stated in the paper. The answers to all other questions raised in the reviewer's point 2 are also answered in the paper.

Response to Reviewers

“Direct observation of deformation and resistance to damage accumulation during shock loading of stabilized nanocrystalline Cu-Ta alloys”

Manuscript NCOMMS-23-30586C-Z

September 1, 2024

We are grateful to Reviewer #4 for carefully reading the revised manuscript as well as the previous response and accompanying documentation. The authors appreciate the reviewer’s complimentary comments regarding the manuscript and its conclusions. This document summarizes our responses to the Reviewer and the changes made in the manuscript. The Reviewers’ comments are reproduced verbatim and colored in blue followed by answers in black. Regarding the reviewer’s comment about ‘presumably with higher resolution figures’. The authors will upload high resolution version of each figure separately to ensure their resolution is maintained for publishing. Finally, the authors have included the highest resolution version of each image within the supplementary file possible without exceeding the file size limit.

Reviewer #4

I have read and reviewed the manuscript titled "Direct observation of deformation and resistance to damage accumulation during shock loading of stabilized nanocrystalline Cu-Ta alloys". The manuscript is well-written, comprehensive, and logical with well-justified conclusions. In the opinion of this reviewer, the manuscript should be accepted as is (presumably with higher resolution figures).

Regarding the reviewer’s comment about ‘presumably with higher resolution figures’. The authors uploaded high resolution version of each figure separately to ensure their resolution is maintained for publishing. Finally, the authors have included the highest resolution version of each image within the supplementary information possible without exceeding the file size limit for a single separate Supplementary Information file.